# Quantifying the Oscillatory Evolution of Simulated Boundary-Layer Cloud Fields Using Gaussian Process Regression

Gunho (Loren) Oh[1] and Philip H. Austin[1]

[1]Department of Earth, Ocean and Atmospheric Sciences, University of British Columbia, Vancouver, BC, Canada

**Correspondence:** Gunho (Loren) Oh (loh@eoas.ubc.ca)

**Abstract.** Average properties of the cloud field, such as cloud size distribution and cloud fraction, have previously been observed to evolve periodically. Identifying this behaviour, however, remains difficult due to the intrinsic variability within the boundary-layer cloud field. We apply Gaussian Process (GP) machine-learning model to the regression of the oscillatory behaviour in the statistical distributions of individual cloud properties. Individual cloud samples are retrieved from high-resolution large-eddy simulation, and the cloud size distribution is modelled based on a power-law fit. We construct the time-series for the slope of the cloud size distribution $b$, a slope that is consistent with satellite observations of marine boundary-layer clouds, by observing the changes in the slope of the modelled cloud size distribution. Then, we build a GP model based on prior assumptions about the cloud field following observational studies: a boundary-layer cloud field goes through a phase of relatively strong convection where large clouds dominate, followed by a phase of relatively weak convection where precipitation leads to formation of cold pools and suppression of convective growth. The GP model successfully identifies the oscillatory behaviour from the noisy time-series, with a period of $95 \pm 3.2$ minutes. Furthermore, we examine the time-series of cloud fraction $f_c$ and average vertical mass flux $\overline{M}$, whose periods were $93 \pm 2.5$ and $93 \pm 3.7$ minutes, respectively. The oscillations reveal the role of precipitation in governing convective activities through recharge-discharge cycles.

## 1 Introduction

Cumulus convection plays a central role in regulating the moisture and energy budget in the atmosphere. However, representing the effect of moist convection remains difficult for general circulation and weather prediction models (Bony and Dufresne, 2005; Bony et al., 2006); for example, the cloud radiative feedback remains the largest source of uncertainty in the estimates of equilibrium climate sensitivity (Ceppi et al., 2017; Mauritsen and Roeckner, 2020; Zelinka et al., 2020), and radiative effects of moist convection remains poorly constrained. These models make widely differing assumptions about the dynamics and thermodynamics of boundary-layer clouds (Ceppi et al., 2017; Lipat et al., 2018; Myers and Norris, 2016) that cannot be directly resolved.

The resolution required to accurately model shallow cumulus convection, on the order of $10\,\mathrm{m}$ in the boundary layer (Sato et al., 2017, 2018), still remains computationally prohibitive (*cf*. Figure 2 in Schneider et al., 2017), as short-term simulations of the global climate with spatial resolutions on the order of a few kilometres have only recently been introduced (Stevens et al.,

2019). As such, the role of high-resolution large-eddy simulation (LES) models in improving our understanding of convective effects continues to be essential.

To improve the indirect representation of the effects of moist convection, large-scale models of the atmosphere must account for the large variability in the dynamics and thermodynamics of the sub-grid scale cloud field. For example, a sub-grid scale radiation scheme that approximates the radiative effects of shallow convection would greatly benefit from having a better

estimate about the geometrical structure and the distribution of clouds, which can be used to improve our estimate of the short-wave radiative effects of low clouds Ceppi et al. (2017). An important aspect of the cloud field in this context is the distribution of cloud sizes (Neggers et al., 2003), which has long been the main topic for observational studies, from aircraft measurements and satellite imagery (Benner and Curry, 1998; Berg and Stull, 2002; Machado et al., 1992; Machado and Rossow, 1993; Plank, 1969; Peters et al., 2009; Raga et al., 1990; Rodts et al., 2003; Wilcox and Ramanathan, 2001; Zhao and Di Girolamo,

2007) to numerical simulations (Brown et al., 2002; Garrett et al., 2018; Neggers et al., 2003) of the cloud field.

Marine boundary-layer clouds have been observed to organize into cellular patterns (Malkus and Riehl, 1964; Nair et al., 1998; Seifert and Heus, 2013) as a response to the formation of cold pools, formed by evaporative cooling due to precipitation (Zuidema et al., 2012). The cold pools promote the formation of negatively buoyant downdrafts that inhibit further growth of thermals (Seifert and Heus, 2013; Seifert et al., 2015; Seigel, 2014a). At the boundaries of these open cells, convective forma-

tion is promoted due to the moistening of downdrafts (Seifert and Heus, 2013) and mechanical lifting due to the convergence of cold pool outflows (Xue et al., 2008).

This dynamics between the formation of cold pools from precipitation and the subsequent formation of clouds has been observed to manifest as temporal oscillations; the cloud field goes through a phase of relatively weak convection, until multiple downdrafts from the cold pools collide into a convergence zone, where convective growth begins anew. For mesoscale

marine boundary-layer stratocumulus clouds, the spatial organization of precipitation is found to be important in promoting subsequent cloud formation and the evolution of open cell convection (Feingold et al., 2010; Koren and Feingold, 2013; Wang and Feingold, 2009; Yamaguchi and Feingold, 2015). High-resolution large-eddy simulations have shown that the formation of cold pools is the main mechanism that drives organized marine stratocumulus convection, which corresponds well to long-term satellite observations (Bretherton and Blossey, 2017; Seifert and Heus, 2013; Zuidema et al., 2012).

The temporal oscillation has also been observed in modelling studies of precipitating cumulus convection (Dagan et al., 2018; Feingold et al., 2017). For both shallow and deep clouds, the dominant mechanism that drives this oscillatory evolution is found to be the formation of cold pools due to evaporative cooling from precipitation in the sub-cloud layer (Seifert and Heus, 2013; Yano and Plant, 2012; Tompkins, 2001). This mechanism is referred to as the recharge-discharge cycle of thermodynamic instability by Dagan et al. (2018), motivated by Bladé and Hartmann (1993), where the evaporative cooling due to precipitation

*charges* instability in the atmosphere, which is *discharged* by convection.

Precipitation facilitates both the spatial organization and temporal oscillation of the cloud field, and is governed by a number of factors including cloud microphysics and cloud layer depth. Aerosols, acting as cloud condensation nuclei (CCN), can influence the cloud microphysics by enhancing the cloud droplet number concentration but suppress droplet growth (Twomey, 1974), and LES studies have shown that when the aerosol concentration is increased, the cloud layer deepens, which then

affects rain formation (Dagan et al., 2017; Seifert et al., 2015). Furthermore, modelling studies have shown that an increase in aerosol concentration influences both the amount and the timing of precipitation; in a polluted environment, the efficiency in precipitation formation is reduced, and as a result, rain formation is suppressed and delayed (Dagan et al., 2018; Seigel, 2014b; Seifert et al., 2015; Yamaguchi et al., 2019).

This study is motivated by these observations of the periodic evolution of the cloud size distribution; if the cloud size distribution evolves periodically over time, the dynamic and thermodynamic properties of the cloud field, such as mass flux and cloud cover, must also oscillate accordingly as the cloud field alternates between the two phases of strong and weak convection. However, numerical algorithms used to detect the period of such oscillation suffer from the presence of noise. Because of that, earlier studies concerning the oscillatory growth of cumulus field relied on manual inspections (Dagan et al., 2018; Koren and Feingold, 2011) or spectral analysis (Feingold et al., 2017) although an accurate estimate of the period of this oscillatory behaviour is yet to be found. Given that, we use the kernel-based machine-learning method using the Gaussian Process (GP) model and apply it to the regression of the periodicity in the observed time-series of the cloud size distribution. GP is a flexible non-parametric Bayesian machine-learning model that is well-suited to be used to infer useful information from a noisy dataset.

GP models have been used for this specific purpose in astronomy (Angus et al., 2017) and medical studies (Cheng et al., 2020; Durrande et al., 2016) to estimate the periodicity in a time-series with observational noise and extract an underlying trend in the observed data. Likewise, we propose the use of a GP model to obtain the underlying trend of a time-series in the presence of observational noise. We use LES model results to sample cloud size densities, construct a cloud size distribution, and follow the evolution of the cloud field properties. Based on the periodic behaviour estimated by the GP model, we re-construct the periodic time-series $f(x)$ without noise, which is compared to the original time-series of cloud size distribution. Lastly, we repeat the GP regression for mass flux and cloud cover to test our hypothesis.

Details of the LES model run is given in Section 2.1. Section 2.2 illustrates how the individual clouds are sampled from the model output. Section 2.3 gives the methods used to construct the time-series of the slope $b$ from the cloud size distribution. Section 2.4 examines the traditional Fourier spectral analysis to identify the oscillatory behaviour within the time-series. A brief introduction of the GP regression method is given in Section 2.5, as well as the construction of kernels, which is further explained in Section 2.6, where we present the method used to estimate the periodicity from the time-series. The fully Bayesian model to estimate the uncertainty in the periodicity estimate is given in Section 2.7. The results are discussed in Section 3 and summarized in Section 4.

## 2 Methods

### 2.1 Model Description

The System for Atmospheric Modeling (SAM; Khairoutdinov and Randall, 2003) version 6.11.8 was used to simulate the CFMIP/GASS Inter-comparison of Large-Eddy and Single-Column Models case (CGILS; Blossey et al., 2013; Zhang et al.,

2013). In this study, we use the large-scale forcing and thermodynamic tendencies of the CGILS S6 regime, representing marine sub-tropical shallow cumulus convection.

The model grid size was set to 25 m in all directions over a 43.2 km $\times$ 12.8 km $\times$ 4.8 km model domain. This elongated, *bowling alley* domain was employed to minimize the effect of periodic boundaries. Because the mean air flows along the elongated axis, most of the convective activity occurs without being spatially recycled as clouds rarely veer from the mean airflow. The temporal resolution of the model was 1 second, and the output was written every minute. We performed a total of 36 hours of simulation, although the first 24 hours were used for the model spin-up and therefore excluded from the analysis, and the last 720 time steps for the remaining 12 hours of simulation were used for the analysis. A long spin-up time was necessary for the boundary layer to reach (quasi-)stability across the domain. The cloud field was then sampled every minute. Details of conditional cloud sampling are presented in the following section.

Doubly periodic domains were employed with a soft top to exclude gravity waves as the possible cause of oscillatory behaviour (Dagan et al., 2018). A two-moment microphysics scheme (Morrison et al., 2005a, b) was used, although the cloud layer remained shallow within 2 km of the boundary layer and no ice formation was observed. The microphysics scheme was initialized with a cloud droplet number concentration (CDNC) value of 120 cm$^{-3}$, which is typical for maritime convection (Rasmussen et al., 2002). Compared to other relevant studies of aerosol-cloud interactions, this represents a relatively pristine environment (Dagan et al., 2018; Seigel, 2014b; Yamaguchi et al., 2019). Other parameters used in the LES model run are based on CGILS S6 control run (Blossey et al., 2013; Tan et al., 2016; Zhang et al., 2013). A diurnally averaged solar insolation was applied, and both short-wave and long-wave radiative effects were calculated using the Rapid Radiative Transfer Model (RRTM) (Clough et al., 2005; Iacono et al., 2008).

To make accurate measurements of vertical mass flux, we implemented the tetrahedral interpolation scheme (Dawe and Austin, 2011). Every time instantaneous cloud fields are being sampled, mass flux rates are integrated over a cloudy surface. This is done by interpolating each grid cell by 48 tetrahedrons, and calculating the changes in the 3-dimensional cloud surface. Recent studies suggest a grid size of roughly 10 m to achieve meaningful statistical accuracy even for shallow convective clouds (Sato et al., 2017, 2018), but the results from the LES model run with 12.5 m grids did not appear to cause significant changes in average vertical mass flux distributions compared to 25 m grids. Since computational resources are limited, and keeping a large domain size (and hence a large number of statistically independent cloud samples) and a longer simulation time is more important for the purpose of this study, we will only examine the results from the 25-m resolution LES run.

Figure 1 shows a three-dimensional snapshot from the LES model run. Clouds are generally scattered over the model domain, and the distribution of smaller clouds appears to be granular. However, areas of vigorous convective activities accompanying precipitation often form in clusters (right side of Figure 1) between areas that are either devoid of clouds or dominated by scattered small clouds (middle of Figure 1). Earlier studies of cloud patterns (Seifert and Heus, 2013; Stevens et al., 2020) suggest such gravel-like patterns could be due to the formation of cold pools. The evaporative cooling due to precipitation can form cold pools, and such patterns can manifest as pronounced convective activities followed by weak, scattered cloudy regime on the leeward side. There are signs of cloud clustering in the simulated cloud field where strongest convective activities (right side of Figure 1, for example) mostly occur in clusters.

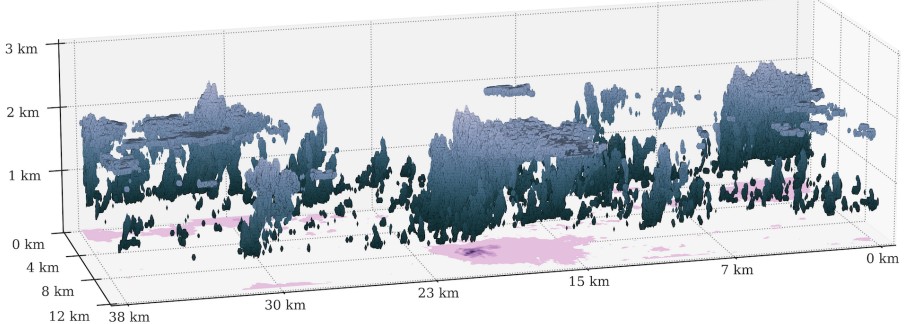

**Figure 1.** A 3-dimensional overview of the LES modelled cloud field from SAM model output, taken 24 hours into the simulation. The shaded regions indicate cloudy cells that contain condensed liquid water. Darker regions are low in altitude, and brighter regions are closer to the cloud top at 2 km. Regions shaded in pink represent areas containing column-integrated precipitable water.

## 2.2 Cloud Sampling

In order to obtain the cloud size distribution, individual clouds need to be sampled conditionally. In this study, horizontally contiguous regions (grid cells) containing condensed liquid water ($q_l > 0$) are considered to be the cloud region. The *size* of a cloud is then defined as the area of the horizontal cross-section, which is the number of grid cells containing condensed liquid water multiplied by the horizontal grid size ($25\,\mathrm{m} \times 25\,\mathrm{m}$). The size distribution contains multiple horizontal cross-sections of the same cloud. However, randomly sampling 60% of the cloud samples did not result in significant changes in the time-series, likely thanks to the sheer number of samples and the shallow nature of the modelled clouds in comparison to the size of the domain, and to the numerical methods used to reduce the statistical uncertainties.

An alternative method by Neggers et al. (2003) is to take a vertical projection of each cloud volume upon a two-dimensional surface. This resembles two-dimensional images taken from a high altitude, which can be useful when comparing the LES output to satellite images, for example. However, the focus of the study is to evaluate the dynamic and thermodynamic properties of the clouds, and simply taking the horizontal cross-section allows us to directly compare the distribution of cloud sizes to that of mass flux. There is another benefit to this approach: when the cloud field is projected onto a two-dimensional surface, the number of smaller clouds sampled from the cloud field increases. This is because smaller clouds are less likely to be projected onto each other. Hence, vertically projected cloud size distribution tends to overestimate the number of smaller clouds, while taking a horizontal cross section (e.g. Brown, 1999) gives a better representation of the realistic three-dimensional cloud size distribution.

Once individual cloud sizes are extracted from the LES output field, we define the *cloud size distribution*. The cloud size distribution $C(a)$ is a cumulative distribution defined as an integral over the *cloud size density* $c(a)$. The cloud size density is a type of probability density function (PDF) that defines the probability of a cloud having a certain size.

Typically, the probability density function is calculated using a histogram with a discrete bin size for a piece-wise estimate, where the density is defined as the frequency of clouds within each discrete *bin*. However, the choice of bin size has a large effect on the resulting distribution, and can yield different results with qualitatively independent features depending arbitrarily on the choice of bin size.

To alleviate these issues, we use the Kernel Density Estimator (KDE; Parzen, 1962a) to reliably estimate the distribution of cloud size density. A kernel density estimator $\hat{k}(x)$ at a point $x$ given a set of $n$ observations $X_1, \ldots, X_n$ is defined as

$$\hat{k}(x) = \frac{1}{nh} \sum_{i=1}^{n} K\left(\frac{x - X_i}{h}\right) \tag{1}$$

which is governed by a *kernel function* $K$ and its *bandwidth* $h$ that control the amount of smoothing applied by the kernel.

We will use a Gaussian kernel $K(x)$ for this purpose, defined as

$$K(x) = \frac{1}{2\pi} e^{-x^2/2} \tag{2}$$

which will be used to smooth the cloud size density function. Each cloud size sample is added to a distribution not as a single point of observation, but a probability distribution based on a Gaussian distribution. It can be considered as an uncertainty in the measurement; that is, each cloud sample is considered to be a Gaussian probability distribution whose width is defined by the bandwidth $h$.

The KDE integrates these probability distributions of cloud sizes, which gives the cloud size distribution $C(a)$. Figure 2 shows the histogram as well as the cloud size distribution based on the KDE using cloud samples taken at 12 hours into the simulation.

The cloud size distribution $C(a)$ in Figure 2 shows a decreasing slope with a scale break for the smallest clouds. The probability density for the smallest clouds is nearly constant. The distribution resembles that of Brown (1999) (*cf*. Figure 12 in Neggers et al., 2003), but with a greater number of smaller clouds. This could be because smaller clouds appear near the cloud base, which is lower than the sampling height used in Brown (1999), reproduced in Neggers et al. (2003). Since the cloud samples in Figure 2 have been taken at all heights, the transition between the smaller and the larger clouds appears to be a lot less abrupt than previously observed.

## 2.3 Cloud Size Model

Given the cloud size distribution $C(a)$, a model of the probability density function needs to be constructed in order to study its temporal evolution. In this paper, we use the power-law distribution (Benner and Curry, 1998; Cahalan and Joseph, 1989; Feingold et al., 2017; Kuo et al., 1993; Neggers et al., 2003; Zhao and Di Girolamo, 2007), which has been widely used to represent the observed distribution of clouds. Here, the cloud size density $c(a)$ is defined as a function of cloud size, or

$$c(a) = c_0 \, a^b \tag{3}$$

where $a$ is the cross-sectional cloud area in m$^2$, and $c_0$ is the coefficient used for the power-law fit. Integrating the cloud size density $c(a)$ over all observed cloud sizes $a$ yields the cloud size distribution $C(a)$.

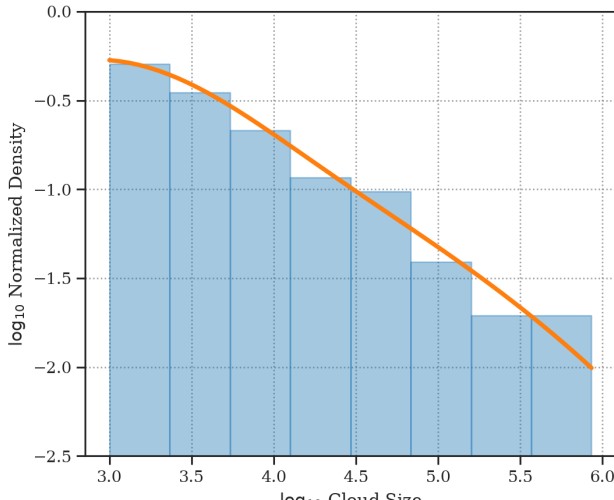

**Figure 2.** A comparison of the histogram (blue) and the kernel density estimate (KDE; orange line) showing $\log_{10}$ of normalized density based on the cloud size distribution over $\log_{10}$ of cloud size in $\text{m}^2$.

From Figure 2, we observe that the cloud size distribution can roughly be divided into two parts, defined by a scale break; the cloud density appears to be relatively constant for the smallest clouds before it decreases linearly. This is a useful feature for the

purpose of this paper. As smaller clouds are short-lived and their contribution to the upward mass flux $M$ is very small, we are interested in the oscillation between the two phases of the cloud field where there are a relative abundance of intermediate-sized clouds, which contributes the most to the mass flux, and where there are a relative abundance of large clouds, which contributes the most to precipitation and formation of cold pools.

We isolate the (quasi-)linear portion by first taking the derivative of the cloud size distribution $C(a)$. A decision tree re-

gression algorithm (Breiman et al., 1984) is used to divide the distribution into two parts by limiting the maximum number of possible branches to two, corresponding to the portion of the distribution with relatively constant slope and the rest of the distribution. This is effectively done by fitting a simple piecewise-constant function $\bar{C}(a)$ to the derivative of $C(a)$, as shown in Figure 3a, where the breakpoint $\hat{a}$ minimizes the error between the distribution $C(a)$ and $\bar{C}(a)$. For this purpose, we make use of the mean square error (MSE) defined as

$$\text{MSE} = \frac{1}{N} \sum_{i=1}^{N} \big( C(a_i) - \bar{C}(a_i) \big)^2 \tag{4}$$

for $N$ samples in the cloud size distribution $C(a)$. As shown in Figure 3a, by minimizing MSE, the decision tree regression algorithm isolates the linear portion of the distribution (green region).

Figure 3b shows how the decision tree splits the distribution based on the derivative of $C(a)$. Once we isolate the relatively linear portion of the distribution, we use the Theil-Sen estimator (Theil, 1950; Sen, 1968) to perform a robust linear regression,

which does well in the presence of outliers and small deviations from the linear trend, as seen in Figure 3a, where the slope

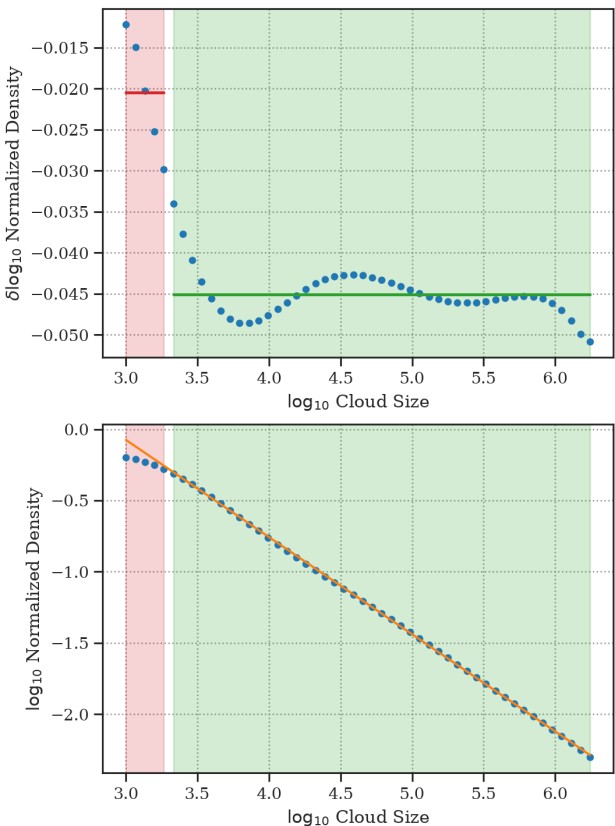

**Figure 3.** The decision tree regression algorithm is applied to the derivative of cloud size distribution (top), which is divided into a region of relatively constant slope (top; green region) and the rest of the distribution (top; red region). The green region is chosen by the algorithm, which is then used to separate the linear portion of $C(a)$ (bottom; green region) from the non-linear portion (bottom; red region). The green region, defined by the part of the distribution with constant slope (bottom; green region), is used to fit a linear curve (bottom; orange line) while the rest of the distribution is ignored (bottom; red region) in order to estimate the slope of cloud size distribution.

represents the ratio between medium-sized and largest clouds. The cloud size distribution $C(a)$ given in Figure 3b represents a normalized probability density function, which will differ from the histograms obtained from observations. Here, the slope is measured to be roughly $b \approx -0.65$. We have calculated $b$ for non-normalized values of $C(a)$, and the time-series is found to vary roughly between $-1.4$ and $-1.7$, which corresponds to a range of $-0.7$ to $-0.85$ based on the method by Neggers et al. (2003), and $-1.7$ to $-1.85$ by Benner and Curry (1998). The measured slopes are slightly smaller in magnitude but comparable to the slope of $b = -1.7$ found in a large-eddy simulation (Neggers et al., 2003) and the slope of $b = -1.98$ from remote sensing observations (Cahalan and Joseph, 1989; Benner and Curry, 1998).

We repeat the calculation of the slope $b$ for 720 time steps, or the entire duration of the simulation excluding the spin-up time. The resulting time-series of the slope $b$ for the cloud size distribution $C(a)$ can be seen in Figure 4. The use of a robust

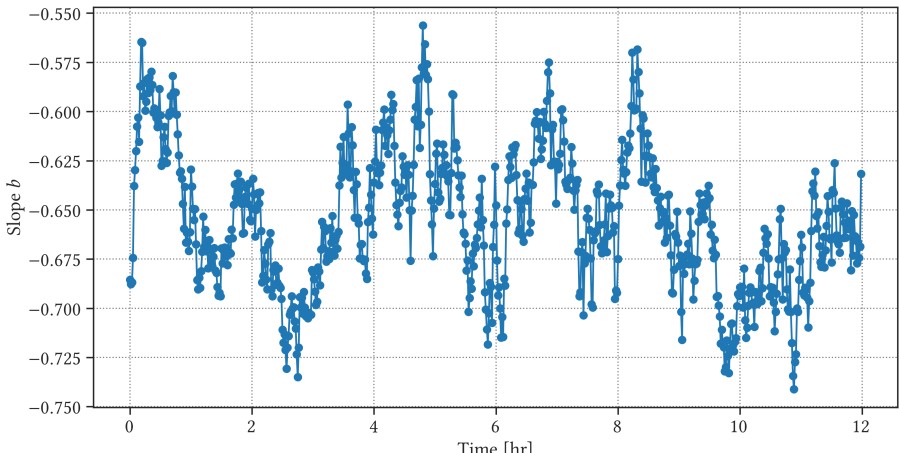

**Figure 4.** The time-series of slope $b$ of the cloud size distribution $C(a)$ for the 12-hour simulation. The slope is calculated every minute (blue line) for the duration of the entire simulation, but the first 9 hours are used for training the GP model.

linear regressor along with a decision tree regression algorithm helps isolate the linear segment of the distribution, which better represents the slope of size distribution of the cloud field and reduces numerical uncertainties involved in the linear regression to obtain the slope $b$.

We are interested in determining whether the fluctuations in the time-series of the cloud size distribution in Figure 4 are consistent with a periodic behaviour. The oscillatory evolution in $b$ is not immediately obvious in Figure 4, and performing an

Augmented Dickey-Fuller (ADF; Dickey and Fuller, 1979; Hamilton, 2020) test shows that the time-series is non-stationary. We would like to quantify the extent to which the time-series is consistent with earlier studies regarding oscillations in the cloud size distribution. In the following section, we follow Feingold et al. (2017) and perform Fourier spectral analysis to identify the underlying periodic behaviour in the observed time-series.

## 2.4 Fourier Spectral Analysis

Given the noisy, non-stationary time-series of the slope $b(t)$ (Figure 4), we tested the traditional spectral analysis based on discrete Fourier transform (DFT) to capture possible oscillatory behaviour (Feingold et al., 2017). Here, given a discrete time-series $f_k = f(k/N)$ for $(k = 0, 1, \ldots, N-1)$, the corresponding DFT, $(\mathcal{F}(f_0), \mathcal{F}(f_1), \ldots, \mathcal{F}(f_{N-1}))$, can be obtained by

$$\mathcal{F}(f_k) = \frac{1}{\sqrt{N}} \sum_{n=0}^{N-1} f(n) \cdot \exp\left(-2\pi i \frac{kn}{N}\right) \tag{5}$$

where $k = 0, 1, \ldots, N-1$.

Equation 5 translates the observed time-series $f_k$ into a function of frequency $k/N$ that is a linear combination of oscillatory, or sinusoidal, components. The strength of each component can then be observed by examining the *power spectral density* of

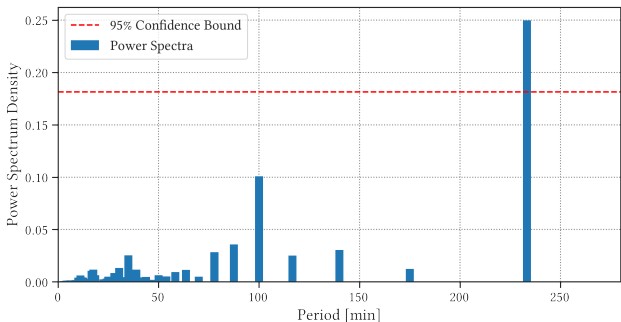

**Figure 5.** The power spectral density (blue) and 95% confidence interval for the time-series $b(t)$ based on the slope of the cloud size distribution $C(a)$, obtained from Fourier spectral analysis.

the time-series. The oscillatory components showing the strongest signals represent the dominant modes of oscillation in the observed time-series.

The power spectral density for a DFT of on a discrete sequence can be estimated by a periodogram, defined as the squared modulus of DFT, or,

$$\mathcal{P}(f_k) = |\mathcal{F}(f_k)|^2 \tag{6}$$

$$= \frac{1}{N} \left( \sum_{n=0}^{N-1} f(n) \cdot \exp\left(-2\pi i \frac{kn}{N}\right) \right)^2 \tag{7}$$

where $k = 0, 1, \ldots, (N-1)/2$ for real-valued input sequence.

Figure 5 shows the estimate of the power spectral density using the periodogram (blue) and the 95% confidence interval (red) obtained from the noisy time-series $b(t)$, plotted as a function of period $T(k) = N/k$. The 95% confidence interval defines the threshold that separates oscillatory signals from noise, against the null hypothesis that all signals in the periodogram are Gaussian noise, and is based on a chi-squared distribution $\chi^2$ with 2 degrees of freedom (Panofsky and Brier, 1958).

There are two prominent periods on the periodogram, one at $T = 100$ minutes and the other at $T = 233$ minutes, but only the latter signal is above the 95% confidence interval. Both signals have periods longer than the 80-minute period observed by Feingold et al. (2017) and no significant signals can be found at shorter periods. The signals are found in the low-frequency regime, and because of that, the frequency bin width is too large to pinpoint the exact period from the periodigram, especially for $T = 233$ minutes. Each coefficient in the periodogram corresponds to period $T(k) = N/k$ for $k = 0, 1, \ldots, N-1$. The longer the period, the coarser the resolution, which greatly reduces the effectiveness of the periodogram in isolating oscillatory components from the noisy time-series. This period is also much longer than the expected oscillatory behaviour (Dagan et al., 2017; Yamaguchi et al., 2019), and is possible that it is a harmonic of a fundamental frequency hidden by noise and non-stationarity.

We have also tested other methods to estimate power spectral density, such as the circular autocorrelation function (ACF; Parzen, 1962b), but (partial) autocorrelation function of the observed time-series $b(t)$ decreases slowly over time, and no

significant lag can be found. The presence of noise as well as the non-stationary nature of the time-series makes it difficult to examine the behaviour of the time-series.

## 2.5 Gaussian Process Regression

As shown in previous section, traditional methods struggle to identify the underlying oscillatory behaviour in the observed time-series. Internal fluctuations in the cloud field and the uncertainties from the numerical methods used in the estimation for the slope of the cloud size distribution can mask the underlying trend in the observation. To handle the uncertainty, we make use of the Gaussian Process (GP) regression (Rasmussen and Williams, 2006). A *Gaussian process* is a set of random variables, any finite number of which have a joint Gaussian distribution. GP models can explicitly model and learn the noise level directly from the data by introducing an explicit noise term that we will use to represent numerical uncertainties involved in sampling and constructing the time-series $b(t)$ of the cloud size distribution.

The time-series can be considered as a regression problem for a latent function $f$ over observations $\mathbf{y} = y(\mathbf{x})$ made at points $\mathbf{x} = x_1, \ldots, x_n$ with noise $\epsilon_y$, or

$$\mathbf{y} = y(\mathbf{x}) = f(\mathbf{x}) + \epsilon_y \,. \tag{8}$$

Assuming no prior knowledge about the noise $\epsilon_y$, we formalize this uncertainty as an additive independent identically-distributed Gaussian distribution with zero mean and a variance of $\sigma_y^2$ (i.e. $\mathcal{N}(0, \sigma_y^2)$). The subscript indicates that the uncertainty comes from the noisy observation $y$.

In this framework, the prior $p(\mathbf{y}|\mathbf{f})$ can be formalized as a (Gaussian) probability distribution, defined uniquely by its mean $\mu(\mathbf{x})$ and a covariance function $k(\mathbf{x_i}, \mathbf{x_j})$:

$$p(\mathbf{y}|\mathbf{f}) \sim \mathcal{N}\big(\mu(\mathbf{x}), \mathbf{K}\big) \tag{9}$$

where $\mathbf{K}$ is a n $\times$ n matrix of covariances ($K_{ij} = k(\mathbf{x_i}, \mathbf{x_j})$) of joint distribution $p(\mathbf{x})$ evaluated at two arbitrary points, which can then be used to define a Bayesian prior that reflects our belief about how the model should behave, prior to observing any data points.

For a set of observations points $\mathbf{x} = x_1, \ldots, x_n$, we can write the covariance matrix $\mathbf{K}(\mathbf{x}, \mathbf{x})$ as

$$\mathbf{K}(\mathbf{x}, \mathbf{x}) = \begin{pmatrix} k(x_1, x_1) & k(x_1, x_2) & \cdots & k(x_1, x_n) \\ k(x_2, x_1) & k(x_2, x_2) & \cdots & k(x_2, x_n) \\ \vdots & \vdots & \ddots & \vdots \\ k(x_n, x_1) & k(x_n x_2) & \cdots & k(x_n, x_n) \end{pmatrix} \tag{10}$$

and because the noise is assumed to be independent, we can modify the covariance function to include the noise term. Because the additive, independent noise term only applies to the diagonal elements of the covariance matrix,

$$\text{cov}(\mathbf{x}, \mathbf{x}) = \mathbf{K}(\mathbf{x}, \mathbf{x}) + \sigma^2 \mathbf{I} \tag{11}$$

where $\sigma^2$ is the variance of our (independent, Gaussian) noise term.

To make predictions, we need to obtain the posterior distribution by conditioning the prior distribution using the observations. To this end, we set up a joint distribution between the observations made at training points $\mathbf{y} = f(\mathbf{x})$ and the function values at testing points $\mathbf{f}_* = f(\mathbf{x}_*)$ as

$$p\left(\begin{bmatrix}\mathbf{y} \\ \mathbf{f}_*\end{bmatrix}\right) = \mathcal{N}\left(\begin{bmatrix}\mu(\mathbf{x}) \\ \mu(\mathbf{x}_*)\end{bmatrix}, \begin{bmatrix}\mathbf{K}(\mathbf{x},\mathbf{x}) + \sigma^2\mathbf{I} & \mathbf{K}(\mathbf{x},\mathbf{x}_*) \\ \mathbf{K}(\mathbf{x}_*,\mathbf{x}) & \mathbf{K}(\mathbf{x}_*,\mathbf{x}_*)\end{bmatrix}\right) \tag{12}$$

which can be conditioned by the observations $\mathbf{y}$ to yield the posterior distribution, which is also Gaussian, as

$$p(\mathbf{f}_*|\mathbf{x}_*,\mathbf{y}) = \mathcal{N}(\mathbb{E}[\mathbf{f}_*], \mathbb{V}[\mathbf{f}_*]) \tag{13}$$

that is fully specified by mean and variance

$$\mathbb{E}[\mathbf{f}_*] = \boldsymbol{\mu}(\mathbf{x}_*) + \mathbf{K}(\mathbf{x}_*,\mathbf{x})\left(\mathbf{K}(\mathbf{x},\mathbf{x}) + \sigma^2\mathbf{I}\right)^{-1}(\mathbf{y} - \boldsymbol{\mu}(\mathbf{x})) \tag{14}$$

$$\mathbb{V}[\mathbf{f}_*] = \mathbf{K}(\mathbf{x}_*,\mathbf{x}_*) - \mathbf{K}(\mathbf{x}_*,\mathbf{x})\left(\mathbf{K}(\mathbf{x},\mathbf{x}) + \sigma^2\mathbf{I}\right)^{-1}\mathbf{K}(\mathbf{x},\mathbf{x}_*) \tag{15}$$

respectively.

A crucial step in Gaussian process regression is determining the covariance $\mathbf{K}$, also called the *kernel*, which embodies the prior knowledge or our assumptions about the observed processes. A kernel function $k(x,x')$ determines how an arbitrary pair of sample points $x$ and $x'$ are related to each other. Essentially, it reflects our belief about how the probability distribution (the target function $f(x)$ under the observed data, as seen in Equation 8) behaves.

The most widely used covariance function is the Square-Exponential (SE), defined as

$$k_{\mathrm{SE}} = \exp\left(-\frac{(x-x')^2}{2\lambda^2}\right) \tag{16}$$

where $\lambda$ is typically defined as a length-scale, or a timescale for the time-series data. The timescale $\lambda$ is one of the hyper-parameters in our GP model that determines how smooth the resulting process varies in time. The SE kernel is most widely used as it gives enough freedom to model a wide range of timescales, and it has an added benefit of being infinitely smooth, which will be useful later.

For the purpose of this study, we will also be using a periodic kernel that assumes a sinusoidal process, which was originally defined by MacKay (1997) as

$$k_{\mathrm{per}} = \exp\left(-\frac{2\sin^2\left(\pi|x-x'|T^{-1}\right)}{\lambda^2}\right) \tag{17}$$

where $\lambda$ is a timescale, and $T$ is the period of oscillation.

The timescale $\lambda$ and the period $T$ are the hyper-parameters, which control the shape of the posterior distribution. These hyper-parameters are initially unknown, although the prior distribution can be specified based on domain knowledge, initial

observations, or some prior assumptions about the data. In order to determine a better estimate of the hyper-parameters, we need to perform inference based on the *marginal likelihood*, which is defined as the integral of the likelihood and the prior, or

$$p(\mathbf{y}|\mathbf{x}) = \int p(\mathbf{y}|\mathbf{f},\mathbf{x})p(\mathbf{f}|\mathbf{x})d\mathbf{f} \tag{18}$$

where the term *marginal* refers to the process of taking an integral over $\mathbf{y}$, or *marginalizing* over the observations, in order to obtain $p(\mathbf{y}|\mathbf{x})$.

Note that under the Gaussian Process model, both the prior and the likelihood must also be Gaussian (MacKay, 1997; Rasmussen and Williams, 2006). The integral above reduces to

$$\log p(\mathbf{y}|\mathbf{x}) = -\frac{1}{2}\mathbf{y}^\top(\mathbf{K}+\sigma^2\mathbf{I})^{-1}\mathbf{y} - \frac{1}{2}\log|\mathbf{K}+\sigma^2\mathbf{I}| - \frac{n}{2}\log 2\pi \tag{19}$$

which defines the marginal log likelihood (MLL). A more detailed derivation can be found in Chapter 2 from Rasmussen and Williams (2006). By maximizing the marginal log likelihood, we can find the hyper-parameters that maximize MLL, which can best explain the observed data.

Based on the time-series of cloud size distribution in Figure 4, we can assume that a simple periodic kernel is not enough to model the complexity of the observed time-series. Therefore, we have added an SE kernel to the periodic kernel to model numerical instability and uncertainty (MacKay, 1997; Rasmussen and Williams, 2006) which we can use to customize our GP model to better reflect the underlying assumptions about the observed dataset. In this case,

$$\hat{k} = k_{\text{SE}} + k_{\text{per}} \tag{20}$$

where the SE kernel has been added to account for the non-stationarity. With no prior assumptions about the underlying dynamics, we have also tested different combinations for $\hat{k}$, such as adding two periodic kernels together in order to model two oscillations at different timescales (Dagan et al., 2018), but using a single periodic kernel for periodicity detection has been proven to be sufficient in our case.

The GP model uses a gradient descent algorithm to find the hyper-parameters that optimize the (log) marginal likelihood. However, there is no guarantee that it will reach the point that maximizes the marginal likelihood, which will define the best set of hyper-parameters. It is possible that even with multiple experiments, the GP model might be fitted to local maxima, as the gradient descent cannot evaluate the full posterior distribution (see Section 2.7 for more information). Fortunately, at least for the timescale in the periodic kernel $k_{\text{per}}$, we can define a reasonable range of values that can represent the oscillation in the cloud size distribution. To demonstrate that the 95-minute period is not a local minimum, we have repeatedly trained our GP model with different initial periods, ranging from 5 minutes to 150 minutes at 5-minute intervals, and all of these tested periods converged to the 95-minute period after training. We have also tested periods longer than 150 minutes, but the resulting posterior distribution ignores most of the variability in the time-series with $T > 150$ minutes. In most cases, the initial period only affected the number of steps that needed to be taken for the gradient descent algorithm to get to the 95-minute period; the further away from the 95-minute period, the longer it took for the algorithm to optimize the periodic kernel.

Given that, we chose an initial period of 90 minutes, which corresponds to the previously observed period of oscillation (Dagan et al., 2018). We have set a lower bound of 15 minutes to timescale $\lambda$ to avoid overfitting and to ensure that most of the

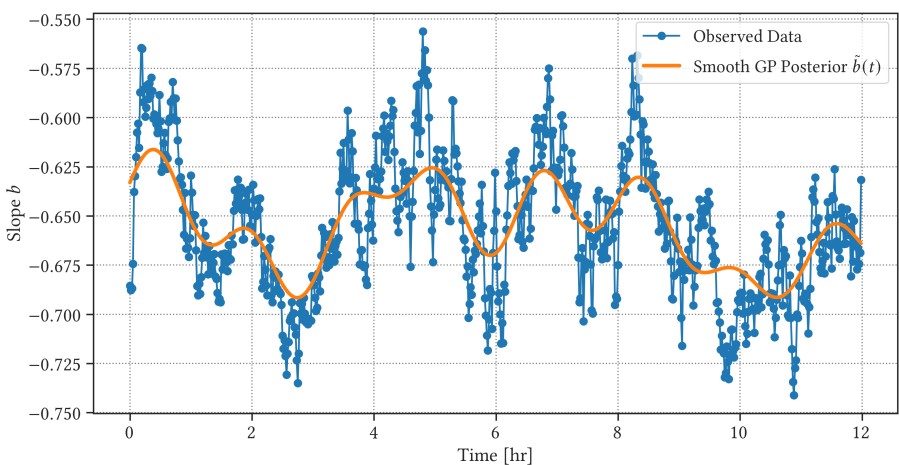

**Figure 6.** Mean posterior of the trained GP model (orange line) compared to the observed slope $b$ of the cloud size distribution $C(a)$ (blue line) for the first 9 hours of simulation used for training.

variability in the observed time-series can be explained by the periodic kernel $k_{\text{per}}$. If the timescale $\lambda$ is allowed to be small, the GP regression model will simply fit the observed data as close as possible regardless of the periodicity.

The GP models are implemented in GPytorch (Gardner et al., 2018) and the hyper-parameters are trained using the Adam optimizer (Kingma and Ba, 2014). The resulting GP posterior distribution with the kernel $\hat{k}$ can be seen in Figure 7. The

335 initial application of the GP model serves two purposes. First, by assuming a smooth variation in the slope $b$, we relegate the uncertainty involved in calculating the slope $b$ from the cloud size distribution $C(a)$ to the noise term. Second, as we assume no prior knowledge about the periodic evolution of $b$, the hyper-parameters can be used as a preliminary estimate for the following procedure, which will be described in the next section.

## 2.6 Periodicity Detection

340 The smoothly modelled distribution $\tilde{b}(t)$ from the mean GP posterior distribution $b(t)$ in Figure 6 corresponds well to the observed time-series, showing the oscillatory behaviour within the noisy observation with a period $T = 95$ minutes. In this particular case, the initial regression attempt yields a good estimate of the hyper-parameters for the observed time-series. However, in situations where a general, long-term trend breaks the quasi-stability assumption, additional steps can to be taken in order to better isolate the oscillatory behaviour of the cloud field, which still remains noisy and non-stationary.

345 The standard practice to account for the non-stationarity is to take the derivative of the time-series $\partial_t \tilde{b}(t) = \partial \tilde{b}(t)/\partial t$. If the oscillation is dominated by a single frequency, the frequency should also characterize the derivative of the oscillation. Given this, we build a GP regression model to estimate the period of the oscillation in $\partial_t \tilde{b}(t)$, which can be seen in Figure 7.

We have found that taking the derivative of $b(t)$ also successfully normalizes the observed time-series, which is useful for statistical analysis. Applying the ADF test (Dickey and Fuller, 1979; Hamilton, 2020) to $\partial_t \tilde{b}(t)$ also confirms that the resulting

time-series is now stationary. As shown in Figure 7, the values of $\partial_t \tilde{b}(t)$ varies with zero mean with no obvious trend over time. There are small variations in the amplitude, but we can now drop the SE kernel to account for the variability in the y-axis, and only use the periodic kernel (i.e. $\hat{k} = k_{\mathrm{per}}$) to estimate of the periodicity $T$. This reduction in the number of hyper-parameters is also necessary to perform Bayesian inference, which will be described in more detail in Section 2.7.

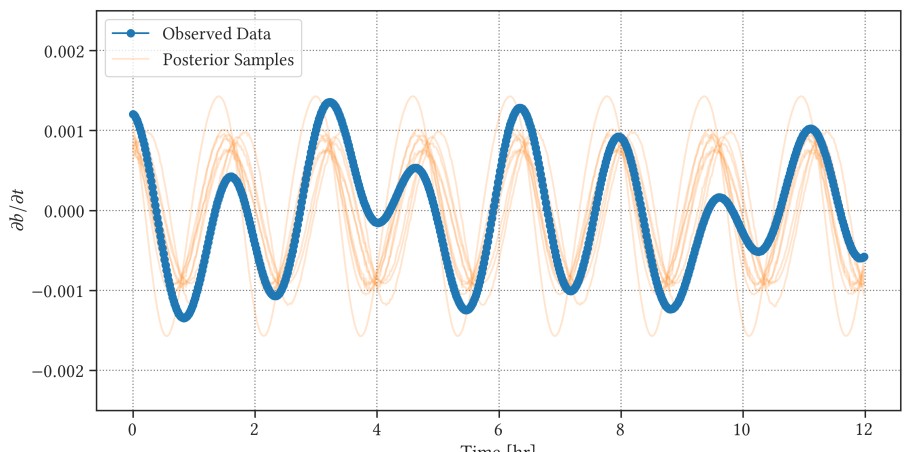

**Figure 7.** Samples from the posterior distribution of the trained GP model (orange line) compared to the observed time-series of $\partial_t \tilde{b}(t)$ (blue line) for the first 9 hours of simulation used for training.

Figure 7 shows the target observation $\partial_t \tilde{b}(t)$ and randomly drawn samples from the posterior distribution. These samples are drawn from the posterior distribution and represent possible realizations of our GP model trained by the observations. The variability in the periodicity seems to be small relative to the uncertainties in the observed time-series.

The results of applying the periodic GP model can be seen in Figure 8, showing the mean posterior distribution for our Gaussian process based on a periodic kernel $k_{\mathrm{per}}$ with noise. Most of the variability comes from deviations in $\partial_t \tilde{b}(t)$, and the use of simple periodic kernel is sufficient to isolate the underlying oscillatory behaviour from the observation. The mean posterior distribution in Figure 8 based on the GP model again yields a period of $T \approx 95$ minutes, which is close to the 90-minute period observed by Dagan et al. (2018) in tropical marine shallow cumulus clouds under precipitating conditions.

Next, we compare the mean posterior distribution from the GP derivative model to $b(t)$. We integrate the mean posterior distribution $\partial_t \tilde{b}(t)$ in Figure 8 to obtain an estimate for $b(t)$ within a constant. For a direct comparison with $b(t)$ we applied the min-max normalization algorithm

$$\tilde{y} = \frac{y - \min(y)}{\max(y) - \min(y)} \tag{21}$$

where $\tilde{y}$ is the normalized time-series, and $y$ is the original observation. This can be used to perform a quick comparison between the integral of the mean posterior distribution $\tilde{b}(t)$ and the observed time-series $b(t)$, which is shown in Figure 9. The normalization, however, makes it more difficult to see that the time-series of the slope $b(t)$ of the cloud size distribution has a negative sign (see Figure 6). A small value in the normalized slope $\tilde{b}$ indicates a more negative slope, or a steeper slope where

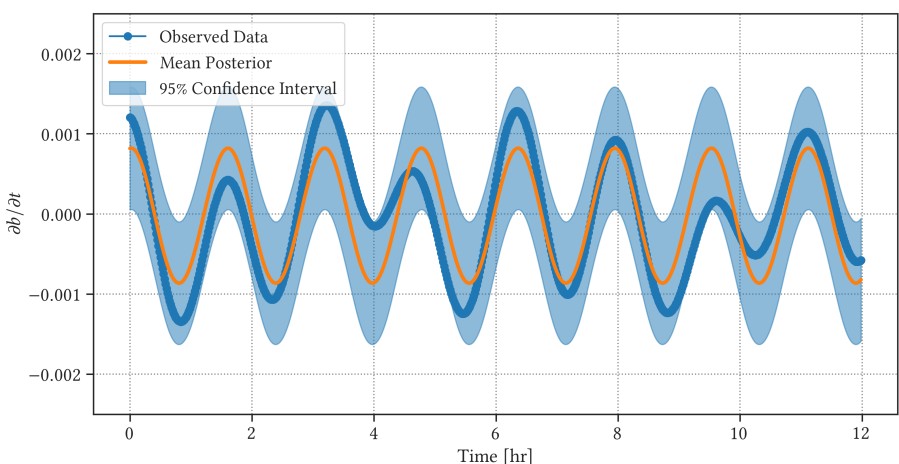

**Figure 8.** Mean posterior distribution of the periodic GP model (orange line) compared to the derivative of time-series $\tilde{b}(t)$ used as the observation (blue line). Shaded regions show (point-wise) 95% confidence interval.

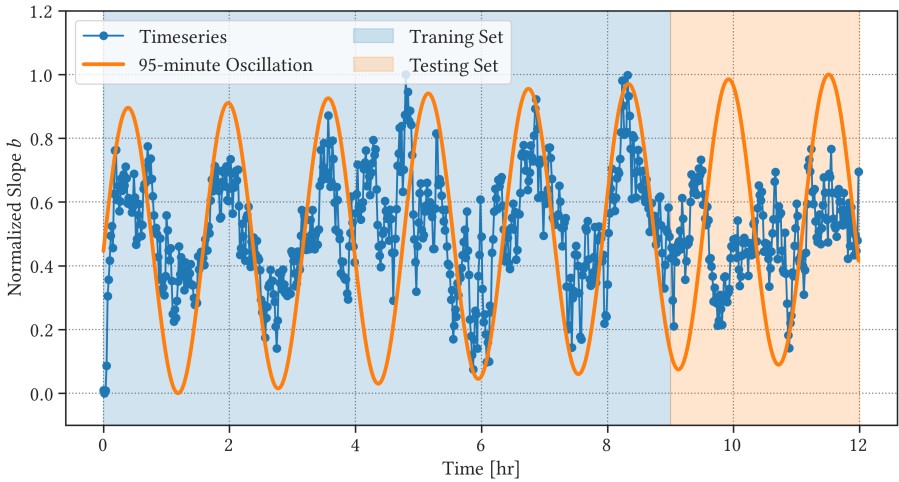

**Figure 9.** Normalized time-series of slope $b$ of the cloud size distribution $C(a)$ (blue line; same as Figure 4), compared to the mean posterior from the periodic GP model (orange line), representing 95-minute oscillation. The full time-series includes the 9-hour training set used for GP model fit (blue region) as well as the 3-hour testing set used for GP model evaluation (orange region).

370     there is a relative abundance of smaller clouds. On the other hand, a large value in $\tilde{b}$ represents a less negative slope where there is a relative abundance of larger clouds.

## 2.7 Fully Bayesian Gaussian Process

While the GP regression model yields posterior distributions with confidence bounds, as shown in Figure 9, we cannot obtain a measure of uncertainty for the hyper-parameters because the gradient descent algorithm cannot evaluate the entire hyper-parameter space. We can instead use a fully Bayesian GP model to estimate the uncertainties in the hyper-parameters of the GP model by stochastically modelling the full posterior distribution. This is done by assuming a prior over model hyper-parameters $\boldsymbol{\theta} \sim p(\boldsymbol{\theta})$, also called the *hyper-prior*; that is, the prior over the hyper-parameters. We can then define joint posterior with the hyper-prior $p(\boldsymbol{\theta})$ as

$$p(\mathbf{f}, \boldsymbol{\theta}|\mathbf{y}) \propto p(\mathbf{y}|\mathbf{f})p(\mathbf{f}|\boldsymbol{\theta}, \mathbf{x})p(\boldsymbol{\theta}) \tag{22}$$

where we have omitted input data $\mathbf{x}$ and $\mathbf{x}_*$ for the sake of simplicity. For a full, illustrative description of Bayesian model selection, refer to Chapter 5.2 in Rasmussen and Williams (2006).

Given the test input data $\mathbf{x}_*$, we retrieve the predictive posterior by integrating the joint posterior

$$p(\mathbf{f}_*|\mathbf{y}) = \iint p(\mathbf{f}_*|\mathbf{f}, \boldsymbol{\theta})p(\mathbf{f}, \boldsymbol{\theta}|\mathbf{y})\,\mathrm{d}\mathbf{f}\,\mathrm{d}\boldsymbol{\theta} \tag{23}$$

$$= \iint p(\mathbf{f}_*|\mathbf{f}, \boldsymbol{\theta})p(\mathbf{f}|\boldsymbol{\theta}, \mathbf{y})p(\boldsymbol{\theta}|\mathbf{y})\,\mathrm{d}\mathbf{f}\,\mathrm{d}\boldsymbol{\theta} \tag{24}$$

whose inner integral reduces to the standard GP posterior, which has the same structure as Equation 14. Using the same formalization in Section 2.5, the outer integral can be estimated as

$$p(\mathbf{f}_*|\mathbf{y}) = \int p(\mathbf{f}_*|\mathbf{y}, \boldsymbol{\theta})p(\boldsymbol{\theta}|\mathbf{y})\,\mathrm{d}\boldsymbol{\theta} \tag{25}$$

$$\simeq \frac{1}{N}\sum_{k=1}^{N} p(\mathbf{f}_*|\mathbf{y}, \boldsymbol{\theta}_k), \tag{26}$$

where $\boldsymbol{\theta}_k \sim p(\boldsymbol{\theta}|\mathbf{y})$. The integral $p(\boldsymbol{\theta}|\mathbf{y}) \sim p(\boldsymbol{\theta}|\mathbf{y})p(\boldsymbol{\theta})$ remains intractable. In order to estimate this integral and obtain the predictive posterior as described in Equation 25, we use a modern variant of Hamiltonian Monte Carlo (HMC) algorithm called *No-U-Turn-Sampler* (NUTS; Hoffman et al., 2014) in Pyro (Bingham et al., 2019).

The fully Bayesian GP model was applied to the time-series of $\partial_t \tilde{b}(t)$. We initially employed normal priors for characteristic timescale $\lambda$ and periodicity $T$ for the periodic kernel, but the GP model quickly converged to the previously observed 95-minute period. To avoid over-fitting, we initialized the Bayesian GP model with an uninformative, uniform prior that allows any quasi-realistic values for the period of oscillation. Unfortunately, the characteristic timescale $\lambda$ fails to converge with the uniform prior in the presence of periodicity, which confirms that the Bayesian GP model can explain the observed time-series solely with a periodic kernel $k_{\mathrm{per}}$, and that the post-processing steps taken in Section 2.6 is useful for the Bayesian inference. The characteristic periodicity $T$ was given a uniform prior $\mathcal{U}(30, 130)$, and Bayesian inference via MCMC has been performed with four chains generating 700 samples, including 300 warm-up samples.

The resulting histogram of the characteristic periods taken from 700 samples can be seen in Figure 10, which shows that the most probable period that can be inferred from the time-series $\partial_t \tilde{b}(t)$ is $95 \pm 3.2$ minutes. It is not surprising that the fully

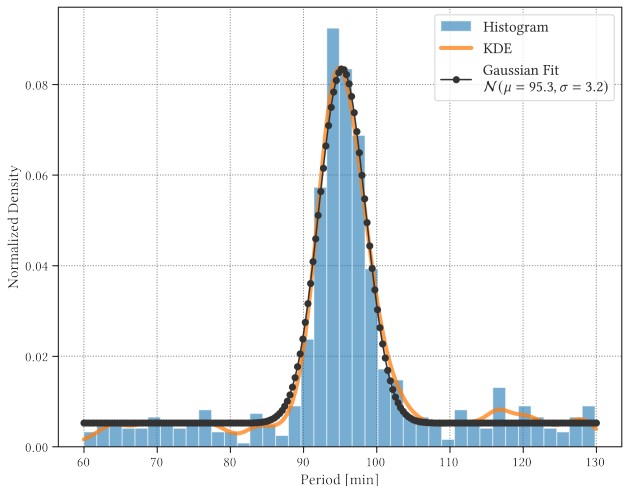

**Figure 10.** A histogram (blue) and the corresponding kernel density estimate (orange) of periods retrieved from Markov chain Monte Carlo (MCMC) experiments. Fitting the Gaussian distribution to KDE (black dots) infers $95 \pm 3.2$ minutes for the observed periodicity in the time-series.

Bayesian GP model converged to the same period from the previous section, and it confirms that non-Bayesian GP model is a reliable framework that can identify oscillatory motions in noisy time-series observations of moist convection.

## 3    Discussion

### 3.1    Cloud Size Distribution

We have estimated the periodicity in the evolution of the slope of the cloud size distribution; the estimated periodicity in $\tilde{b}(t)$ is $T = 95 \pm 3.2$ minutes, which corresponds well to previous studies involving marine shallow cumulus clouds (Dagan et al., 2018). This periodicity is slightly longer than 80 minutes estimated by a Fourier spectral analysis of an LES cloud field (Feingold et al., 2017). This could be due to the differences in precipitation strength, aerosol concentration, or domain size. In the bowling-alley domain, clouds are allowed to develop along the direction of the mean wind, which corresponds to the elongated axis of 42 km. Given that, we expect the development of the cloud field to resemble the larger model run by Dagan et al. (2018). Also, the lower-frequency mode of oscillation found in these studies makes it difficult to accurately determine the periodicity, as described in Section 2.4.

The simulated time-series of the mean posterior distribution for $\tilde{b}(t)$ generally corresponds well to the observed changes in $b(t)$, except at 4 and 10 hours from the beginning of the time-series. Small-scale fluctuations within the cloud domain could disrupt large-scale convective and precipitative patterns, even with a large model domain. To gain more insight about the internal variations within the cloud field, we have visualized the three-dimensional cloud field for the duration of the simulation. Figure 11 shows a snapshot of the modelled cloud field taken at 250 minutes into the simulation.

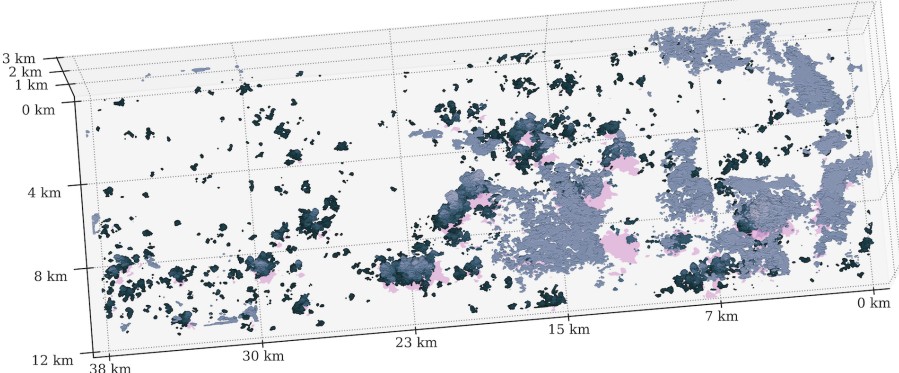

**Figure 11.** Same as Figure 1, but seen from the top of the atmosphere at 250 minutes into the simulation.

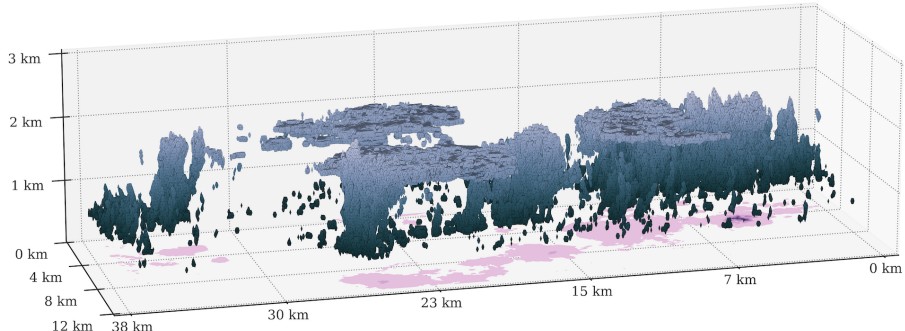

**Figure 12.** Same as Figure 1, taken at 600 minutes into the simulation.

Based on Figure 9 at around $t = 4$ hours, the cloud field is expected to go towards a phase of relatively weak convection
where there is a relative abundance of smaller clouds. Small normalized values correspond to more negative slopes of the cloud
size distribution. However, upon inspecting the evolution of the cloud field during this time, large structures form at the top of
the cloud layer (light grey regions in Figure 11) as a result of strong convective activity, which persist until $t = 5$ hours into the
simulation. Once the large, thin layer of clouds at the top dissipates, the deviation the observed time-series of $b$ becomes much
smaller. During this time, as shown in Figure 11, the cloud field is dominated by the growth of small clouds. However, a thin
layer of clouds persists and skews the slope $b$ of the cloud size distribution.

On the other hand, around 9 hours into the simulation, the cloud field is expected to go towards a phase of relatively strong
convection where there is a relative abundance of larger clouds. Visually inspecting the development of the cloud field (Figure
12) suggests that strong convective activities occur in groups, and clouds tend to merge and become much larger, which reduces
the relative number of large clouds compared to small ones that are less likely to merge with other clouds or reach the cloud

top layer. Because of this, the normalized value of $b$ of the cloud size distribution becomes smaller, indicating a much steeper slope despite the strong convective activity.

As shown in Figure 12, clouds form in groups and merge into large convective columns. The successive formation of clouds can also be seen, where clouds appear in a line following the advection of the large clouds (Moser and Lasher-Trapp, 2017). This can also be attributed to the effect of spatial organization, where the cloud field in Figure 12 can be divided into regions

of strong convection, followed by regions devoid of clouds. The development of large clouds can be seen, as expected by our GP model, but these clouds could have been merged into small regions due to spatial organization of the cloud field. Features of organized shallow convection have been observed in large-eddy simulations with small to moderate domain sizes (Seifert and Heus, 2013; Xue et al., 2008). However, such features are sporadic and the clouds become scattered again at $t = 11$ hours into the simulation.

We were not able to reliably isolate a high-frequency oscillation reported in previous studies, either for $T \approx 10$ minutes (Dagan et al., 2018) or for $T \approx 15$ minutes (Feingold et al., 2017) with a modified prior distribution. This is likely due to the large noise in the time-series $b(t)$ (*cf.* Figure 4), as well as small variations in the amplitude of the oscillation. Over-fitting of noisy time-series becomes an issue in such cases as the GP model quickly adheres to random noise, or observational uncertainties, but not necessarily to high-frequency oscillation.

Given that the primary mode of oscillation found in this study has a period of 95 minutes, we suspect that the size of the domain used for the LES run (43.2 km × 12.8 km) might have made it more difficult to isolate the evolution of individual clouds (Dagan et al., 2018), as we expect the 15-minute period to be correlated to the convective timescale for individual clouds, rather than changes in the mean cloud field (Feingold et al., 2017; Heus et al., 2009). It is also possible that such convective oscillation exists, but due to the nature of non-linear oscillation at small scales, our attempt in resolving the time-series with a

periodic kernel was not able to adequately capture the complex dynamics of non-linear oscillators (Koren and Feingold, 2011; Koren et al., 2017; Seifert and Heus, 2013) at the scale of individual clouds.

Lastly, using fully Bayesian GP model to estimate the uncertainty in the characteristic periodicity $T$ (Figure 10) confirms that without any prior knowledge, except with an assumption that a reasonable period value would lie somewhere between 30 and 130 minutes, we could reliably retrieve a time-series oscillation that is statistically significant, with a period of $T = 95 \pm 3.2$

minutes.

## 3.2 Cloud fraction

As the oscillation is assumed to be driven mainly by spatial and temporal organization of precipitating clouds, we make an assumption that the same periodic evolution in the dynamic and thermodynamic properties of the mean cloud field, such as cloud cover $f_c$ (Feingold et al., 2017) and vertical mass flux $M$, can also be observed. Given that, we applied the GP regression

method for the cloud fraction over the model domain $f_c$, which is simply the fraction of the domain that is covered by clouds. Cloud fraction is calculated by projecting the 3-dimensional clouds onto the ground, and calculating the fraction of the 2-dimensional grid cells that are covered by cloudy region (with condensed liquid water, or $q_l > 0$). We repeat the calculation every minute to construct the cloud fraction time-series $f_c(t)$.

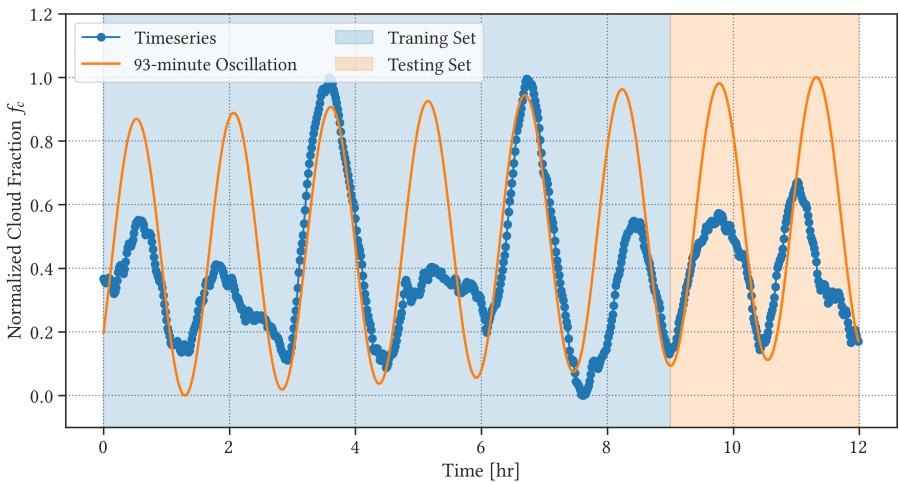

**Figure 13.** Same as Figure 9, but for cloud fraction $f_c$.

We applied the same numerical techniques presented in Section 2, including the uncertainty estimate for the hyper-parameters
using the fully Bayesian GP model (Section 2.7). The results of applying the GP regression method can be seen in Figure 13, where the period of oscillation estimated by the fully Bayesian GP model is shown to be $T_f = 93 \pm 3.7$ minutes.

The period lies well within the range of estimated periods for $\tilde{b}(t)$, even when the modelled oscillatory evolution $\tilde{b}(t)$ deviates from the observed values of $b(t)$. This is because merging and splitting of clouds do not significantly change the area that is covered by those clouds. Likewise, persistent clouds forming at the top of the cloud layer tend to form over shallower clouds
near the bottom of the cloud layer, and they do not seem to affect the oscillatory evolution in $f_c$.

The proposed GP regression model can accurately estimate the periodic evolution of cloud fraction, but cloud fraction $f_c$ cannot account for internal variations in the cloud size distribution. The estimated mean posterior distributions of $b$ and $f_c$ are well correlated, but the deviations noted in Section 3.1 are hidden in the time-series of $f_c$. Large values of $f_c$ are normally associated with a relative abundance of large clouds, but the actual state of the cloud size distribution could differ; for example,
the cloud field can be dominated by small clouds near the cloud base but covered by a thin layer of clouds at the top (Figure 11), in which case the radiative properties of the cloud field will be different from one that is dominated by large clouds, despite both having large cloud fraction.

### 3.3 Average Cloud Vertical Mass Flux

We have also applied the GP regression method to vertical mass flux $M$, which is defined at each grid cell as

$$M = \rho w \mathcal{A} \tag{27}$$

where $\rho$ is air density [$\mathrm{kg\,m^3}$], $w$ is vertical velocity of the air [$\mathrm{m\,s^{-1}}$], and $\mathcal{A}$ is the activity field which is 1 for the cloudy cell (containing condensed liquid water, or $q_l > 0$), and 0 otherwise.

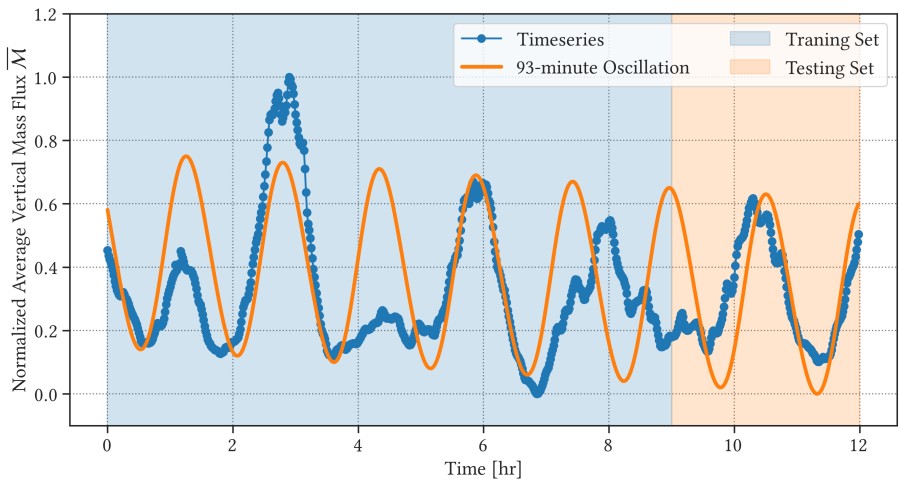

**Figure 14.** Same as Figure 9, but for average vertical mass flux $\overline{M}(t)$.

The average mass flux $\overline{M}(t)$ can then be obtained by calculating average mass flux across the cloud field and taking the average value over the vertical column as the vertical distribution of mass flux remains relatively consistent within the cloud layer. We use this value as a proxy for convective vigour in the modelled cloud field. We calculate $\overline{M}$ at each time step to construct the time-series of average mass flux $\overline{M}(t)$ for the simulated 12-hour period (Figure 14).

For vertical mass-flux over the mean cloud field, the period is estimated to be $T_M = 93 \pm 2.5$ minutes, which agrees well with the estimated ranges for both $\tilde{b}$ and $f_c$. Figure 14 shows both the normalized mass flux time-series $\overline{M}(t)$ as well as the integrated mean posterior of the periodic GP model. The time-series of $\overline{M}(t)$ tends to be much smoother than that of the cloud size distribution because the fluxes have been averaged over the model domain.

The mean posterior distribution shown in Figure 14 follows the observed time-series quite closely for the training set, except around $t = 4$ hours and $t = 9$ hours into the simulation. The suppression of vertical mass flux around $t = 4$ hours is connected to the deviation in $\tilde{b}$ (Figure 9). Small clouds form across the model domain during this period, which corresponds to large values of average vertical mass flux $\overline{M}(t)$, but the a thin layer of clouds persists near the top and reduces overall values of $\overline{M}(t)$. On the other hand, the oscillatory behaviour in $\overline{M}(t)$ is disrupted between $t = 7$ and $t = 10$ hours of simulation. As noted in Section 3.1, spatial organization of clouds could influence the deviation; during this time, clouds tend to form close to existing convective columns, which can be seen as regions of strong convective activities followed by regions devoid of clouds (*cf.* Figure 12). The formation of cold pools from convective precipitation will result in a convergence of descending moist air, which can disrupt the oscillatory evolution of the cloud field and force the formation of large, organized groups of clouds.

In both cases, it would be useful to diagnose the changes in the vertical distribution of mass flux $M$ to examine the factors contributing to internal fluctuations and gain more insight into how vertical distributions of moisture and momentum periodically re-arrange over time. This also allows us to examine how the clouds mix in terms of the entrainment and detrainment

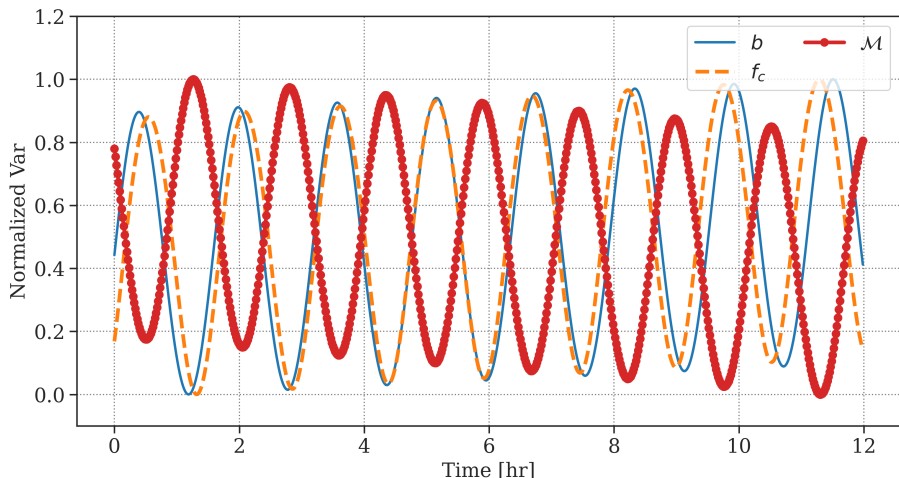

**Figure 15.** Normalized time-series of slope $\tilde{b}$ of the cloud size distribution $C(a)$ (blue), average vertical mass flux $\overline{M}(t)$ (red) cloud fraction $f_c$ (orange), based on the periodic GP model.

rates, where

$$\frac{\partial}{\partial z}M = e - d \tag{28}$$

for rates of entrainment $e$ and detrainment $d$ [$\mathrm{kg\,m^{-2}\,s^{-1}}$], but a preliminary examination of domain-average vertical mass flux profiles reveals that the variability within the cloud field is too large and more work needs to be done, which is outside the scope of this study.

### 3.4 Precipitation Flux

As shown in Figures 13 and 14, oscillations in $\tilde{b}$ and $f_c$ are well-correlated, while the oscillation in $\overline{M}$ seems to be lagging.
That is, the peaks in $\overline{M}$ correspond roughly to the troughs in both $\tilde{b}$ and $f_c$. To visualize the relationship, we plotted normalized mean posterior distributions from our GP model for the three variables in Figure 15.

Figure 15 shows the oscillatory evolutions of the mean posterior distributions for $\tilde{b}$, $f_c$ and $\overline{M}$, which gives more insight about how the marine boundary-layer cumulus clouds evolve over time in a high-resolution model. When there is a relative abundance of larger clouds, the normalized slope $\tilde{b}$ of the cloud size distribution and cloud fraction $f_c$ become larger, which
corresponds to a less negative (less steep) slope $b$ of the cloud size distribution. Hence, the changes in cloud fraction $f_c$ is correlated to the number of large clouds; that is, the number of large clouds (mostly observed as anvil-like structures near the cloud layer top) determines how much of the model domain is covered by clouds.

On the other hand, peaks in mass flux $\overline{M}$, averaged across the model domain, is associated with smaller values of $\tilde{b}$. This corresponds to a more negative (steeper) slope $b$ of the cloud size distribution, which suggests a relative abundance of smaller
clouds, associated with smaller values of cloud fraction $f_c$. The formation of smaller clouds is connected to an average increase

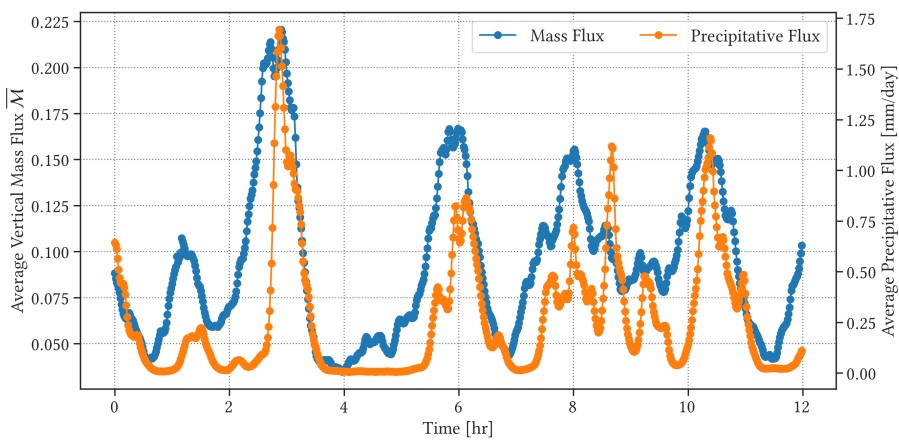

**Figure 16.** Normalized time-series of average vertical mass flux $\overline{M}(t)$ [kg m$^{-2}$ s$^{-1}$] and precipitation flux [mm per day].

in vertical mass flux across the domain, while the formation of larger clouds is associated with weak overall mass flux. This is indicative of the recharge-discharge mechanism, where strong convective activities are associated with precipitation and the formation of cold pools, which suppresses cloud formation. As the larger clouds dissipate and surface fluxes introduce atmospheric instability, average vertical mass flux across the model domain increases and small clouds develop, which then promotes the formation of larger clouds (Dagan et al., 2018). The increase in average vertical mass flux can also be attributed to the dynamic lifting at the boundaries of cold pools, where descending moist air from cold pools merge to promote the growth of large clouds.

To further examine the relationship between the strength of convection, inferred from average mass flux $\overline{M}$, and precipitation, we plotted the time-series of the average vertical mass flux $\overline{M}$ compared to the precipitation flux, averaged over the cloud layer in Figure 16. It is clear that periods of strong vertical mass flux are followed by vigorous precipitation, where latent heat gets released, warms the cloud layer, and promotes the formation of large clouds. As large clouds form and begin to precipitate, the sub-cloud layer cools due to evaporation, and warming of the cloud layer and cooling of the sub-cloud layer reduce (discharge) atmospheric instability within the boundary layer, rapidly weakening convection and precipitation until surfaces fluxes re-introduce atmospheric instability.

Hence, the time-series of mass flux $\overline{M}$ and precipitation in Figure 16 are consistent with the proposed recharge-discharge cycle (Dagan et al., 2018), which is primarily driven by precipitation and latent heat release due to convection. That is, each recharge-discharge cycle consists of a 95-minute oscillation where atmospheric instability is *charged* by surface fluxes and *discharged* by convection/precipitation over the 95-minute period. This period, however, can be disrupted by internal fluctuations and spatial organization of the cloud field (*cf.* Section 3.1). Furthermore, changes in aerosol concentrations will affect precipitation efficiency, where rain formation in polluted environments tends to be less efficient and therefore slower (Seigel, 2014b; Yamaguchi et al., 2019).

To isolate the role of precipitation in driving the proposed recharge-discharge cycle, we performed a follow-up simulation of the CGILS case where rain formation in the cloud microphysics scheme is artificially suppressed. Likewise, we have tested the GP regression method to the mass flux time-series in shallow convection during BOMEX (Holland and Rasmusson, 1973) case with precipitation turned off as well. In both cases, the GP regression method described in Section 2 failed to converge towards a single periodicity and no prominent oscillatory behaviour could be found.

## 4 Conclusions

We performed a high-resolution, large-eddy simulation of tropical marine boundary-layer clouds and implemented numerical methods to analyze the temporal changes in the slope $b$ of the cloud size distribution $C(a)$ that reflects the state of moist convection across the modelled cloud field. LES modelled boundary-layer clouds were conditionally sampled, and the probability distribution of cloud sizes was defined by the Kernel Density Estimator (KDE; Parzen, 1962a). The slope $b$ of the cloud size distribution at each time step was then obtained by a decision tree algorithm and a robust linear regression method. By applying these numerical methods for every model output, we constructed the time-series of the changes in the slope $b(t)$ for the cloud size distribution, which can be used as a proxy of the state of the modelled cloud field.

The numerical steps described in this study are taken primarily to reduce noise and numerical uncertainties involved in calculating the slope $b$. We have also tested the cloud size distributions of vertically projected three-dimensional cloud field (Neggers et al., 2003; Feingold et al., 2017) using the histogram, but this resulted in a very noisy time-series and an unsuccessful attempt at estimating the underlying oscillation (*e.g.* Figure 9 in Feingold et al., 2017). We have also examined other criteria from the literature, but obtaining a stable time-series for the slope $b$ remains difficult.

Under the assumption that the noisy time-series $b(t)$ consists of a single periodic oscillation with large observational variability (Equation 20), we retrieve the underlying trend from the observation using a GP regression model, which is then used to estimate the period of oscillation by a GP model with a periodic kernel $k_{\mathrm{per}}$. The kernel defines our *a priori* belief of the underlying behaviour, and the GP model optimizes the hyper-parameters, especially the period $T$ of the oscillation, that maximizes the likelihood of the posterior distribution against the observations. We further calculate the uncertainty of the estimated periodicity using a fully Bayesian GP inference. The smooth gradient $\partial_t \tilde{b}(t)$ of the slope of the cloud size distribution $C(a)$ is used as a proxy to determine the underlying behaviour of cloud size distribution, whose period is estimated to be $T = 95 \pm 3.2$ minutes.

We have also applied this technique to total cloud cover $f_c$ and average vertical mass flux $\overline{M}$. Using the fully Bayesian GP model, we identified $T_M = 93 \pm 2.5$ minutes as the period of oscillation for average vertical mass flux, and $T_f = 93 \pm 3.7$ minutes for total cloud cover. Dynamic properties of the cloud field, therefore, are found to oscillate along with the cloud size distribution. The estimated periods agree well with the study involving a satellite observation of open cells (Koren and Feingold, 2013). There are other studies where the cloud field properties can be seen to show similar oscillatory behaviour, which can be identified by the GP regression method presented here; for example, the time-series of cloud cover and liquid-

water path (LWP) (Figure 2 from Seifert and Heus, 2013) as well as mass flux (Figure 4 from Plant and Craig, 2008) also appear to oscillate in time (Figure 4).

The oscillation in the time-series of the precipitation flux (Figure 16) is also consistent with the 95-minute period found for the cloud size distribution and the mass flux, suggesting that the oscillation is primarily driven by the recharge-discharge cycle in atmospheric instability (Dagan et al., 2018). The atmospheric instability is weakened (discharged) when clouds grow large and begin to precipitate, as the upper boundary layer warms due to latent heat release and the lower layer cools due to evaporation. This stabilization of the boundary layer continues until cloud growth slows down and precipitation stops. After that, surface fluxes destabilize the boundary-layer from below and convective phase starts again due to atmospheric instability. This cycle takes roughly 95 minutes in the LES model run based on the CGILS S6 case. It should also be emphasized that the same periodic behaviour can be observed using the two-moment microphysics scheme, whereas the bin microphysics scheme has been used by Dagan et al. (2018). The thermodynamic processes that govern the recharge-discharge cycle seem to work consistently in multiple simulations of the boundary-layer atmosphere.

The oscillatory behaviour of precipitating marine boundary-layer clouds has been noted in the literature (Dagan et al., 2018; Feingold et al., 2017; Seigel, 2014b; Yamaguchi et al., 2019). Given that the LES model run here represents sub-tropical shallow cumulus regime, it is not surprising that the 95-minute oscillation found by our GP model is consistent with previous modelling studies of boundary-layer clouds. We have initialized the model run with a pristine atmosphere (where CDNC was set to $120 \, \mathrm{cm^{-3}}$), but studies have shown that changing the aerosol concentration can influence the precipitation efficiency, and therefore the period of oscillation (Dagan et al., 2018; Yamaguchi et al., 2019). As the atmosphere becomes more polluted, we expect this periodicity to increase. We have also observed the spatial organization of shallow convective clouds that disrupts the oscillatory evolution (Figure 12), which can also be seen in studies modelling the boundary-layer cloud field over a smaller domain (Wang and Feingold, 2009). Reducing the size of the model domain may reduce the effect of spatial organization and make it easier to estimate the periodic behaviour of the cloud field using traditional methods. However, given that these factors can manifest even on a smaller domain, a robust method to estimate the periodicity of a noisy, non-stationary time-series is still useful, especially if no smallest, *optimal* domain size exists where the recharge-discharge cycle can be isolated.

Lastly, examining the changes in the vertical distribution of mass flux $M$ can give more insight into how the cloud field mixes with the environment over time, especially because we have already examined the temporal changes in $f_c$, or $\partial \sigma / \partial t$ where $f_c = \sigma$ (Section 2.6). Given the results from the high-resolution large-eddy simulation, the mass continuity of the cloud field can be examined in terms of entrainment and detrainment rates, or

$$\rho \frac{\partial \sigma}{\partial t} + \frac{\partial M}{\partial z} = e - d \tag{29}$$

for vertical mass flux $M = \rho w \sigma$ and rates of entrainment $e$ and detrainment $d$ $[\mathrm{kg \, m^{-2} \, s^{-1}}]$. We have observed that both cloud fraction and vertical mass flux oscillate over time, but the latter can also be seen to be lagging by half a period (Section 3.4). Given that, investigating how the vertical distribution of mass flux changes over time can provide valuable insights about the dynamics of shallow convection.

*Code and data availability.* The SAM LES model is publicly available at http://rossby.msrc.sunysb.edu/SAM.html, and the exact set of parameters for the model run, and the model output including conditionally sampled cloud sizes can be found at https://zenodo.org/doi/10.5281/zenodo.11355002 (Oh, 2024b). Jupyter notebooks showing the detailed steps of the numerical analysis are also available at https://zenodo.org/doi/10.5281/zenodo.11356150 (Oh, 2024a).


*Author contributions.* LO designed and ran the LES model run, carried out the numerical analysis and prepared this manuscript. PHA conceptualized this project and reviewed the manuscript.

*Competing interests.* The authors declare that they have no conflict of interest.

*Acknowledgements.* This work was partially supported by Korea Polar Research Institute (KOPRI) grant funded by the Ministry of Oceans and Fisheries (KOPRI PE24010). We would like to thank KOPRI for kindly providing the computational resources required to perform various machine-learning tasks for this paper.

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
