# Peer review of "Quantifying the Oscillatory Evolution of Simulated Boundary-Layer Cloud Fields Using Gaussian Process Regression"

_EGUsphere, 2024_

## Referee Comment (RC1)

Review: Quantifying the Oscillatory Evolution of Simulated Boundary-Layer Cloud Fields Using Gaussian Process Regression

By Oh and Austin

This manuscript addresses the question of oscillatory behavior in the cloud systems. It emanates from earlier work that identified oscillations in cloud size distributions in open cell stratocumulus and shallow non-precipitating cumulus (BOMEX). It develops very nice, refined tools to differentiate the periodicity of the oscillations in noisy time-series data. Key results are that the authors find a similar periodicity in the slope of the cloud size distribution, the mass flux, and the cloud fraction (~ 93 min). The cloud size distribution oscillation is anticorrelated with the mass flux oscillation. The mass flux oscillation precedes the cloud fraction oscillation by half a period. The authors attribute the oscillations to charge-discharge cycles driven by cloud development and rain, respectively.

**Main comments**

The methodology developed by the authors is very rigorous and I appreciated the effort that went into finding methods to discern oscillatory signals in noisy cloud model data. I do feel however that the authors are missing some important context, and that they could have put more effort into developing a deeper understanding of what the oscillations are telling us. In my mind the paper could place more focus on interpretation. I recognize that this is a GMD manuscript and therefore that some of my suggestions may not seem pertinent. Perhaps the authors are planning a deeper analysis of the oscillations in a different paper. Nevertheless, I provide these comments as food for thought.

1) Quite a few studies that have discussed oscillations in cloud systems are referred to. These range from shallow cumulus (BOMEX) to the deeper CGILS case discussed here to open cell stratocumulus. I don't think enough distinction is made between these cases. For example, an open-cell stratocumulus system is characterized by significant internal coupling through colliding outflows associated with surface precipitation such that clear oscillatory behavior is expected. The 90 min periodicity in those systems is likely a time required for spatial rearrangement of the up- and down-drafts (or charging vs discharging areas). Shallow BOMEX clouds barely precipitate at the surface and are in a different class so that arguments about cloud-rain charge-discharge don't seem relevant, and certainly the fact that the signals are weak is to be expected. The CGILS S6 case precipitates more significantly and is quite different from BOMEX. Another study of precipitating Cu (10.1029/2019JD031073) shows that aerosols can change the charge-discharge time depending on the degree of clustering (e.g., Fig. 10).
The paper would really benefit from a more nuanced discussion that discriminates between cloud types, cloud organization, precip amounts, and coupling in the cloud system. Note, I thinks this is important even for a GMD publication.

2) The imbalance between the more technical parts of the paper and the interesting discussion that starts on pg 16 could be corrected a bit. I felt that there were missed opportunities on the discussion to dig into the boundary layer physics. Examples: lines

412-414, lines 415-417, but I think there is much more that can be said in Sections 3.2, 3.3, and 3,4.

The figures require some work since one has to mentally superimpose plots of b and M, or pick off peaks and troughs in different plots to see that they are out of phase. Also, because b is normalized, it should be made clearer that a smaller b is a more negative b, i.e. a larger fraction of small clouds. The plots contain important information but reference to them is too cryptic in my opinion.

3) What about the possibility of a charge (production of instability) – discharge (consumption of instability) as a driver of oscillations. https://doi.org/10.5194/acp-5-2749-2005. Is this how non-precipitating systems differ from precipitating systems?

4) Oscillations in M lead the oscillations in CF by 45-50 minutes. This made me wonder which size clouds contribute most to CF, which of course depends on b. I also wondered how detrainment at cloud top contributes to this (see Fig. 1).  Could you strengthen this analysis and discussion?

5) Dagan et al. (2018) change their domain size and show that oscillations get smoothed out for larger domain sizes – at least for BOMEX. Have you tested the sensitivity of the periodicity to domain size. One could imagine that small domains introduce higher frequency oscillations because of the periodic boundary conditions. If the oscillations become harder and harder to discern at large domain size, are they really important? The fact that they can be discerned by your GP regression is very nice but they may not have much physical significance in a natural cloud system unless the system experiences strong internal coupling (e.g., open-cell Sc).

**Other:**

6)  It would be nice to know how the normalized b values translate to actual b values that e.g., a satellite imager would see.

7) Line 132: could you help with physical meaning of 'bandwidth h'?

8) Line 102, $q_l > 0$ is a very low threshold, unless your cloud edges are very sharp. Does $q_l > 0.01$ g/kg change the picture?

---

## Referee Comment (RC2)

Review of Oh and Austin 2024:

The goal of this paper is to analyze the periodicity of vigorous versus quiescent periods in a trade cumulus region (CGILS S6). The slope of the distribution of large-cloud sizes, the convective mass flux, the cloud fraction, and the surface precipitation rate are all analyzed and the lead/lag relationship between these variables is used to hypothesize about the underlying physics.

I thought the paper was generally well written and the topic is interesting, but I have some concerns about the methodology.

Major concerns:

1. I'm skeptical of your Gaussian process modeling (GPM).
    a. I see spectral analysis as the gold standard for identifying periodic signals in a dataset - if the dataset has a signal at a certain wavelength, it *must* show up in a spectral analysis. I also disagree that spectral analysis is particularly sensitive to noise - noise usually shows up in a power spectrum as a lower limit to spectral power across a range of frequencies. It is true that this lower limit can be so large that it swallows your signal, but there are 2 peaks in Fig. 5 that rise above this floor. In this context it is hard to square the results of your GPM against what I see in your power spectrum or even via visual inspection of your time series.
    b. I'm pretty sure all you are doing is effectively band-pass filtering the data by looking only for periods between 5 and 150 minutes which throws out the large signal at ~230 minutes and causes the secondary peak at ~100 minutes in Fig 5 to rise to the level of statistical significance because so much of the rest of the power spectrum has been thrown away. I could be wrong about this, but I would need to see my hypothesis proven wrong before I'd be comfortable accepting this paper.
    c. I think my simple story in 3b is getting lost in a maelstrom of math. Einstein once said that "a model should be as simple as possible, but no simpler". In that context, why do you perform your analysis on the derivative of the signal rather than the signal itself (when there is no trend in the data to get rid of and subtracting the linear regression would be simpler if there were)? Why do you need GPM instead of applying a digital filter? I'd like to see clear justification for each step towards complexity you take.
2. I'm concerned that oscillations in cloud populations might be fairly local in nature such that averaging over a 45 km fetch averages out most of the variability you seek. Thankfully, you can easily address this question (which would be a nice addition to your paper)! In particular, can you repeat your analysis by only analyzing output over subsets of your domain? If so, does the peak frequency or spectral power change as a function of region size? Also, discussion of averaging over a large region occurs around

line 330, but I'd like to see more discussion and I'd like to see it show up in the experimental design section.

3. When I look at Fig 9, I see ~4 very clear waves with 95 min period while the rest of the timeseries doesn't fit this frequency very well at all. This leads me to suggest that your active/quiet hypothesis works well for some chunks of your simulation but something is interrupting the signal for other chunks. Can you watch a movie of your simulation or perform other analysis to figure out what is interrupting your behavior of interest? Finding conditions which disrupt these oscillations would be a very noteworthy contribution!

4. I'm also worried that the narrow dimension of your bowling-alley domain might be corrupting your results. It seems to me that the narrow horizontal direction would constrain the size of cloud radii. And since your whole paper centers around the cloud size distribution, locking the biggest clouds to the domain size seems like it could affect your conclusions. Can you reassure me that this isn't happening? Just providing citations of previous work showing this isn't a concern would be fine.

5. around line 105: I don't understand whether you're just taking cloud cross sections at a single level (e.g. 1 km) or whether you consider each vertical level to be an independent cloud. The latter approach could be problematic because clouds at different levels wouldn't be independent samples which would mess up statistical analysis tests. This would also distort interpretation because what we think of as a distribution of cloud sizes could in fact just represent the vertical structure of a single cloud. Can you clarify what you're doing and address potential concerns if you are using all levels?

6. There's a lot of focus on matching the periodicity found in previous studies, but do we know that such periodicity should be spatially and temporally invariant? Recharge/discharge seems like it would be proportional to boundary layer depth and the vigor of turbulent/convective mixing. Timescales might also be different in the different sorts of cloud analyzed by these previous papers.

Minor points:
1. It would be nice to see the timeseries of cloud size distribution, mass flux, and cloud fraction all on a single graphic. The degree of correlation between variables is difficult to see by flipping between graphics. A lead/lag correlation analysis would be a nice way to show the relationships between these fields.

2. A cartoon explaining the relationship between cloud size, mass flux, cloud fraction, and precipitation in terms of hypothesized regional life cycle might be useful for driving home what you're seeing.

3. I really like Figure 1! It does a nice job of grounding the paper in reality. It might be even more powerful if you showed snapshots like this for both a very active and very quiescent period.

4. The KDE gives you a non-parametric PDF which you can use to get the slope you need… so why do you introduce a power-law distribution? I suspect all you want is to compute the linear regression of the nonparametric KDE PDF in log-log space (after chopping off the nonlinear bit on the left-hand side of the curve in e.g. Fig 3). I suspect I'm complaining about your wording rather than what you've actually done.

5.  I found Fig 3 to be a bit confusing. I would expect the derivative of the original plot to come below/after the plot it is the derivative of. It also took me a while to see the delta in front of the ylabel in panel a, which added to my confusion. Further, it would be reassuring to see several snapshots in time to make sure your log-log linearity assumption is justified in general instead of just in this test case. Alternatively, you could plot RMSE of fit (or something like that) for your linear regression as a function of time.

Typos:
1. L86 - I think "cell by 48" should be "cell INTO 48"
2. L221-222: you already said that noise is included above.
3. L262L you call timescale l here when it should be \lambda.
4. L503: "base" should be "based"

---

## Author Comment (AC2)

**Response to comments for "Quantifying the Oscillatory Evolution of Simulated Boundary-Layer Cloud Fields Using Gaussian Process Regression"**

Loren Oh*

June 25th, 2024

We appreciate the constructive feedback for the submitted manuscript.

**Responses to Comments**

1. Quite a few studies that have discussed oscillations in cloud systems are referred to. These range from shallow cumulus (BOMEX) to the deeper CGILS case discussed here to open cell stratocumulus. I don't think enough distinction is made between these cases. For example, an open-cell stratocumulus system is characterized by significant internal coupling through colliding outflows associated with surface precipitation such that clear oscillatory behaviour is expected. The 90 min periodicity in those systems is likely a time required for spatial rearrangement of the up- and down-drafts (or charging vs discharging areas). Shallow BOMEX clouds barely precipitate at the surface and are in a different class so that arguments about cloud-rain charge-discharge don't seem relevant, and certainly the fact that the signals are weak is to be expected. The CGILS S6 case precipitates more significantly and is quite different from BOMEX. Another study of precipitating Cu (10.1029/2019JD031073) shows that aerosols can change the charge-discharge time depending on the degree of clustering (e.g., Fig. 10). The paper would really benefit from a more nuanced discussion that discriminates between cloud types, cloud organization, precipitation amounts, and coupling in the cloud system. Note, I thinks this is important even for a GMD publication.

As the reviewer noted, this manuscript is primarily meant to be a technical paper, where the focus is in introducing the GP regrssion method with Bayesian inference, and in confirming
* * *
*Corresponding author: loh@eoas.ubc.ca

that the estimated oscillation corresponds well to earlier studies. Still, we agree that we can expand on the discussion of physical mechanisms driving the observed oscillatory behaviour within the LES modelled cloud field. The revised manuscript will include a new LES simulation of the CGILS S6 case with precipitation turned off, and it will allow us to directly analyze the role of precipitation in driving the oscillatory behaviour.

We should also mention that we have in fact not observed the 90-minute period in the shallow BOMEX case. As the reviewer suspected, shallow clouds for the BOMEX case barely precipitate, and the oscillatory behaviour cannot be driven by precipitation. This was an error on our end, and we apologize for the confusion. We have performed the analysis again for all the cases discussed in the manuscript, and the revised manuscript will include a discussion on the oscillatory behaviours of different cloud fields.

> 2. The imbalance between the more technical parts of the paper and the interesting discussion that starts on pg 16 could be corrected a bit. I felt that there were missed opportunities on the discussion to dig into the boundary layer physics. Examples: lines 412-414, lines 415-417, but I think there is much more that can be said in Sections 3.2, 3.3, and 3,4. The figures require some work since one has to mentally superimpose plots of b and M, or pick off peaks and troughs in different plots to see that they are out of phase. Also, because b is normalized, it should be made clearer that a smaller b is a more negative b, i.e. a larger fraction of small clouds. The plots contain important information but reference to them is too cryptic in my opinion.

We have come up with an additional figure that shows how the oscillations occur in different phases (see Figure 1 here), which will be added to the revised manuscript. Figure 1 makes it clear that the average vertical mass flux $\mathcal{M}$ varies inversely to the slope of the cloud size distribution. The reviewer is also correct that the time-series is normalized and the slope of the cloud size distribution has a negative sign. We will ensure that this is clear in the revised manuscript.

> 3. What about the possibility of a charge (production of instability) - discharge (consumption of instability) as a driver of oscillations. Is this how non-precipitating systems differ from precipitating systems?

I am somewhat confused about this comment. Did you mean convective organization instead of the charge-discharge cycle, which is discussed in detail? If so, the revised manuscript will include an extended discussion based on the results from the non-precipitating CGILS S6 case. We believe that given the oscillatory behaviour in cloud fraction, convective organiza-

[Figure]

Figure 1: Mean posterior distributions from the GP regression for the slope $b$ of the cloud size distribution (blue), cloud fraction $f_c$ (orange) and average vertical mass flux $\mathcal{M}$.

tion will certainly play a role in the oscillatory behaviour of the cloud field.

> 4. Oscillations in M lead the oscillations in CF by 45-50 minutes. This made me wonder which size clouds contribute most to CF, which of course depends on b. I also wondered how detrainment at cloud top contributes to this (see Fig. 1). Could you strengthen this analysis and discussion?

Because we ignore the smallest clouds in the estimation of the slope of the cloud size distribution (see Figure 3 in the submitted manuscript), medium-sized clouds contribute the most to both cloud fraction and mass flux. Figure 1 also makes it easier to see this. Cloud-top detrainment is directly responsible for the formation of anvil-like structure near the top of the cloud layer, as the reviewer noted. As you can see in Figure 1, $f_c$ is in phase with $b$, which indicates that cloud-top detrainment of large clouds corresponds to the peak in cloud fraction. Figure 1 will be added to the revised manuscript, and will allow us to expand the discussion on the physical mechanisms behind the oscillatory behaviour.

> 5. Dagan et al. (2018) change their domain size and show that oscillations get smoothed out for larger domain sizes - at least for BOMEX. Have you tested the sensitivity of the periodicity to domain size. One could imagine that small domains introduce higher frequency oscillations because of the periodic boundary conditions. If the oscillations become harder and harder to discern at large domain size, are they really important? The fact that they can be discerned by your GP regression is very nice but they may not have much physical significance

in a natural cloud system unless the system experiences strong internal coupling (e.g., open-cell Sc)

That is exactly the merit of this manuscript, we'd say; that is, although the oscillations become more difficult with increasing domain size, numerical methods can be used to identify the oscillatory behaviour and estimate the uncertainties involved using Bayesian inference. Feingold et al. (2017) also did this on a smaller domain, but had difficulties in identifying the period in the cloud size distribution. It becomes more and more difficult to identify the oscillatory behaviour due to the variability in the time-series as the domain size becomes larger. So the effect of the oscillatory behaviour is simply hidden in a large cloud field, and the variability in the properties of the cloud field can be better determined if we can account for the oscillatory evolution.

6. It would be nice to know how the normalized b values translate to actual b values that e.g., a satellite imager would see.

We have added a paragraph on the range of $b$ based on the posterior distribution in Figure 9 of the submitted manuscript.

7. Line 132: could you help with physical meaning of 'bandwidth h'?

This has to do with translating a point measurement into a probability distribution (which we assume to be Gaussian). The bandwidth h is the width of this probability distribution, representing the uncertainty in the measurement. This concept comes up again in the GP regression method, and we will make its physical significance clearer in the revised manuscript.

8. Line 102, $q_l > 0$ is a very low threshold, unless your cloud edges are very sharp. Does $q_l > 0.01$ g/kg change the picture?

Definitely not for the cases being presented in the manuscript. We have tested different conditional criteria for cloud sampling in the past, but have not found any differences in statistical distributions.

---

## Author Comment (AC3)

**Response to comments for "Quantifying the Oscillatory Evolution of Simulated Boundary-Layer Cloud Fields Using Gaussian Process Regression"**

Loren Oh*

June 25th, 2024

We appreciate the constructive feedback for the submitted manuscript.

**Responses to Comments**

**Major Points**

1a. I see spectral analysis as the gold standard for identifying periodic signals in a dataset - if the dataset has a signal at a certain wavelength, it *must* show up in a spectral analysis. I also disagree that spectral analysis is particularly sensitive to noise - noise usually shows up in a power spectrum as a lower limit to spectral power across a range of frequencies. It is true that this lower limit can be so large that it swallows your signal, but there are 2 peaks in Fig. 5 that rise above this floor. In this context it is hard to square the results of your GPM against what I see in your power spectrum or even via visual inspection of your time series.

We agree that a true signal will show up in a spectral analysis, but we do not agree that noise would always behave so nicely, especially since we are dealing with a very noisy time-series where the noise occurs at different scales. Here, we defined the *noise floor* to be the magnitude of the spectral power for Gaussian noise using a chi-squared distribution. We have revised the manuscript to include more details about this particular choice. The goal here was to see if the traditional spectral analysis could identify the 90-minute period with enough confidence (i.e. 95% confidence that the signal is not white noise). From what I
* * *
*Corresponding author: loh@eoas.ubc.ca

[Figure]

Figure 1: Power spectral density from FFT, with periods larger than 200 minutes filtered out.

can tell from the spectral analysis, the time-series indeed contains large variability for the spectral analysis to be effective (more on this later).

> 1b. I'm pretty sure all you are doing is effectively band-pass filtering the data by looking only for periods between 5 and 150 minutes which throws out the large signal at 230 minutes and causes the secondary peak at 100 minutes in Fig 5 to rise to the level of statistical significance because so much of the rest of the power spectrum has been thrown away. I could be wrong about this, but I would need to see my hypothesis proven wrong before I'd be comfortable accepting this paper.

We understand the concern, but the reviewer is conflating band-pass filtering for spectral analysis and gradient descent algorithm used by the GP regression method. First of all, band-pass filtering the Fourier series still does not yield a strong enough signal given the noise and/or the variability within the periodicity (see Figure 1).

The choice of the prior (periodicity) in the GP regression method, on the other hand, is not meant to be deterministic; the only reason we repeated the regression over a range of periods is to ensure that there are no local minima with steep gradient. In most cases the GP model converged nicely to the 90-minute periodicity. Furthermore, there are reasons to be skeptical of the 230-minute period – it is too long for either the lifetime of shallow clouds or cloud organization, it could be a harmonic of higher-frequency oscillations that are hidden in the noise, and the frequency bin of the DFT is too large at this scale.

[Figure]

Figure 2: Normalized time-series of slope b of the cloud size distribution C(a) (blue line), compared to the the mean posterior from the periodic GP model (orange line), representing the 238-minute oscillation.

We can definitely use our GP model to see what the low-frequency oscillation looks like. We used a 230-minute period as the prior, and the model converged to an oscillation with a period of 238 minutes. See Figure 2; it is clear that the longer period ignores much of the measured time-series.

Still, we do need to acknowledge that the spectral analysis *did* find a signal at 90 minutes. We have added more details on this matter.

> 1c. I think my simple story in 3b is getting lost in a maelstrom of math. Einstein once said that "a model should be as simple as possible, but no simpler". In that context, why do you perform your analysis on the derivative of the signal rather than the signal itself (when there is no trend in the data to get rid of and subtracting the linear regression would be simpler if there were)? Why do you need GPM instead of applying a digital filter? I'd like to see clear justification for each step towards complexity you take

I think the reviewer meant to say point 1b (correct me if I am wrong).

Taking the derivative for the GP regression serves two purposes. First, as we have noted in the manuscript, it detrends the time-series, and not just a linear trend; oftentimes the trend is difficult to remove by a linear regression. Second, it normalizes/standardizes the time-series, which makes the GP model more stable. It is not particularly necessary for this case, and we do obtain the same result without taking the derivative (albeit with more

steps), but we would say it is advisable to take the derivative of a noisy time-series.

We admit that some details have been left out when we were trying to reduce the length of the manuscript for submission. We have added these details to the steps we have taken for the GP regrssion model.

> 2. I'm concerned that oscillations in cloud populations might be fairly local in nature such that averaging over a 45 km fetch averages out most of the variability you seek. Thankfully, you can easily address this question (which would be a nice addition to your paper)! In particular, can you repeat your analysis by only analyzing output over subsets of your domain? If so, does the peak frequency or spectral power change as a function of region size? Also, discussion of averaging over a large region occurs around line 330, but I'd like to see more discussion and I'd like to see it show up in the experimental design section.

We performed the analysis again over a (randomly chosen) quarter of the domain. Figure 3 shows the resulting power spectral density. As the reviewer suspected, the signals appear stronger on a smaller domain, which is expected as expanding the size of the domain would introduce more variability in our time-series. Still, the spectral analysis returned multiple signals. We can see additional signals at 144 and 180 minutes above the 95% confidence bound, which is interesting.

The signals at 180 and 240 minutes are likely harmonics of higher-frequency oscillations, and the bin size for the DFT is too large to determine what the actual period should be. Given that, we used 90 and 144 minutes as priors for our GP model, and tested how accurate they were (see Figures 4 and 5). The former converged to a 95-minute period, which is expected, and the latter converged to a 144-minute period, which is less accurate. So it does not become significantly easier to identify the oscillatory evolution of the cloud field even on a smaller domain, as the variability is still too large for traditional methods.

We will also expand our discussion on the effect of domain size.

> 3. When I look at Fig 9, I see 4 very clear waves with 95 min period while the rest of the timeseries doesn't fit this frequency very well at all. This leads me to suggest that your active/quiet hypothesis works well for some chunks of your simulation but something is interrupting the signal for other chunks. Can you watch a movie of your simulation or perform other analysis to figure out what is interrupting your behavior of interest? Finding conditions which disrupt these oscillations would be a very noteworthy contribution!

[Figure]

Figure 3: Power spectral density from FFT for a quarter of the domain.

We have definitely looked at the snapshots to see what is interrupting the oscillatory behaviour. At around 10 hours into the sampling period, multiple columns of clouds merge to form a large anvil-like structure near the top of the cloud layer. It is not clear if this is due to convective organization, as based on a visual inspection, clouds seem to simply follow the mean wind (along the longer side of the domain). We revised the manuscript to include more details about these observations.

> 4. I'm also worried that the narrow dimension of your bowling-alley domain might be corrupting your results. It seems to me that the narrow horizontal direction would constrain the size of cloud radii. And since your whole paper centers around the cloud size distribution, locking the biggest clouds to the domain size seems like it could affect your conclusions. Can you reassure me that this isn't happening? Just providing citations of previous work showing this isn't a concern would be fine.

It is not. The width (or the shorter end) of the bowling-alley domain 12.8 km, which would be long enough for a square domain (12.8 km × 12.8 km). The width of the largest cloud is only roughly 1 km or so. Moreover, because of the way the LES model is set up (no large-scale forcing, etc), clouds will grow towards the size of the domain given the right conditions, in which case we would know that the size of the domain is not large enough. The simulation presented here showed no signs of that happening. There are no previous studies specifically for the CGILS S6 case, but doi.org/10.5194/gmd-6-1261-2013 should give you a good idea. This is based on a BOMEX case, and size distributions behave similarly in the two cases.

[Figure]

Figure 4: Normalized time-series of slope b of the cloud size distribution C(a) (blue line), compared to the the mean posterior from the periodic GP model (orange line), representing the 95-minute oscillation.

    5. around line 105: I don't understand whether you're just taking cloud cross sections at a single level (e.g. 1 km) or whether you consider each vertical level to be an independent cloud. The latter approach could be problematic because clouds at different levels wouldn't be independent samples which would mess up statistical analysis tests. This would also distort interpretation because what we think of as a distribution of cloud sizes could in fact just represent the vertical structure of a single cloud. Can you clarify what you're doing and address potential concerns if you are using all levels?

As it is described in line 103, we take the horizontal cross-sections of all cloudy regions. We also sample 60% of these cloud samples to reduce the number of cloud samples (horizontal cross-sections) that are connected vertically. This detail must have been removed when we were trying to shorten the manuscript (as it was too long before the submission). Still, sampling all horizontal cross-sections does not change our result, and just makes the variability in the time-series somewhat larger.

    6. There's a lot of focus on matching the periodicity found in previous studies, but do we know that such periodicity should be spatially and temporally invariant? Recharge/discharge seems like it would be proportional to boundary layer depth and the vigor of turbulent/convective mixing. Timescales might also be different in the different sorts of cloud analyzed by these previous papers.

[Figure]

Figure 5: Normalized time-series of slope b of the cloud size distribution C(a) (blue line), compared to the the mean posterior from the periodic GP model (orange line), representing the 144-minute oscillation.

We tried to make no prior assumptions about the nature of the oscillatory behaviour, and we simply found the 90-minute periodicity to be consistent with previous studies.

We are working on a follow-up paper on the effect of the properties of the cloud field on this oscillatory behaviour, but we will add a discussion on possible factors that can affect this oscillation.

**Minor Points**

> 1. It would be nice to see the timeseries of cloud size distribution, mass flux, and cloud fraction all on a single graphic. The degree of correlation between variables is difficult to see by flipping between graphics. A lead/lag correlation analysis would be a nice way to show the relationships between these fields.

Multiple people pointed this out, so to address this point we have come up with an additional figure (see Figure 6 here) showing the oscillations in the three variables.

> 2. A cartoon explaining the relationship between cloud size, mass flux, cloud fraction, and precipitation in terms of hypothesized regional life cycle might be useful for driving home what you're seeing.

We will work on this, but it might be more appropriate for a follow-up paper looking at the physical mechanisms of the oscillatory behaviour.

[Figure]

Figure 6: Mean posterior distributions from the GP regression for the slope $b$ of the cloud size distribution (blue), cloud fraction $f_c$ (orange) and average vertical mass flux $\mathcal{M}$.

     3. I really like Figure 1! It does a nice job of grounding the paper in reality. It might be even more powerful if you showed snapshots like this for both a very active and very quiescent period.

Thank you. We definitely thought of that, and do show more snapshots when we present this particular research in person, but the manuscript was already too long.

     4. The KDE gives you a non-parametric PDF which you can use to get the slope you need...so why do you introduce a power-law distribution? I suspect all you want is to compute the linear regression of the nonparametric KDE PDF in log-log space (after chopping off the nonlinear bit on the left-hand side of the curve in e.g. Fig 3). I suspect I'm complaining about your wording rather than what you've actually done.

This definitely has to do with inertia (i.e. traditional literature) and we do not need it once we use KDE. I will clean up the section and make it clear that those are two different things.

     5. I found Fig 3 to be a bit confusing. I would expect the derivative of the original plot to come below/after the plot it is the derivative of. It also took me a while to see the delta in front of the ylabel in panel a, which added to my confusion. Further, it would be reassuring to see several snapshots in time to make sure your log-log linearity assumption is justified in general instead of just in this test case. Alternatively, you could plot RMSE of fit (or something like

that) for your linear regression as a function of time.

This might have to do with a typo in the paragraph that introduces Figure 3. We will re-write this paragraph to ensure the steps taken to obtain the figure. We will also calculate the RMSE for the linear regression when we calculate the slopes of the cloud size distributions.

**Typos**

Thanks for catching these. We will fix them in the revised manuscript.

---

## Author Comment (AC4)

**Response to comments for "Quantifying the Oscillatory Evolution of Simulated Boundary-Layer Cloud Fields Using Gaussian Process Regression"**

Loren Oh*

February 12, 2025

We sincerely appreciate the constructive feedback for the submitted manuscript. We have made extensive changes to the manuscript based on the reviews, and performed additional model runs to reflect the points raised by the reviewers. The following document summarizes those points, and the changes we've made.

The line numbers in the responses refer to those in the revised manuscript.

**Responses to Comments**

1. Quite a few studies that have discussed oscillations in cloud systems are referred to. These range from shallow cumulus (BOMEX) to the deeper CGILS case discussed here to open cell stratocumulus. I don't think enough distinction is made between these cases. For example, an open-cell stratocumulus system is characterized by significant internal coupling through colliding outflows associated with surface precipitation such that clear oscillatory behaviour is expected. The 90 min periodicity in those systems is likely a time required for spatial rearrangement of the up- and down-drafts (or charging vs discharging areas). Shallow BOMEX clouds barely precipitate at the surface and are in a different class so that arguments about cloud-rain charge-discharge don't seem relevant, and certainly the fact that the signals are weak is to be expected. The CGILS S6 case precipitates more significantly and is quite different from BOMEX. Another study of precipitating Cu (10.1029/2019JD031073) shows that aerosols can change the charge-discharge time depending on the degree of clustering (e.g., Fig.
* * *
*Corresponding author: loh@eoas.ubc.ca

10). The paper would really benefit from a more nuanced discussion that discriminates between cloud types, cloud organization, precipitation amounts, and coupling in the cloud system. Note, I thinks this is important even for a GMD publication.

We must first apologize for the confusion concerning the BOMEX case. There were two errors in the draft caused by an unintentional mix-up. The original analysis included in the draft was actually done on a CGILS case on a small domain, not the BOMEX case. The BOMEX case we meant to analyze also had precipitation turned off, and upon performing the correct analysis, we found no oscillatory evolution in the cloud field as expected.

The CGILS S6 case used for this study is not meant to represent deeper convection, but strictly shallow convection with more precipitation. We have added more details about the CGILS case; the relevant paragraph at line 90 now reads: "The System for Atmospheric Modeling (SAM; Khairoutdinov and Randall, 2003) version 6.11.8 was used to simulate CGILS (CFMIP/GASS Inter-comparison of Large-Eddy and Single-Column Models) case (Blossey et al., 2013; Zhang et al., 2013). In this study, we use the large-scale forcing and thermodynamic tendencies of the *CGILS S6 regime*, representing marine sub-tropical shallow cumulus convection". The clouds in the CGILS S6 regime used in this study do not appear to be significantly different from the ones in BOMEX (see also Bretherton and Blossey, 2017; https://doi.org/10.1002/2017MS000981), except that the precipitation seems to be more vigorous.

We do, however, agree that the manuscript is rather light on the discussion of the literature. We have replaced the paragraph starting at line 36, which now reads: "Marine boundary-layer clouds have been observed to organize into cellular patterns (Malkus and Riehl, 1964; Nair et al., 1998; Seifert and Heus, 2013) as a response to the formation of cold pools, formed by evaporative cooling due to precipitation (Zuidema et al., 2012). The cold pools promote the formation of negatively buoyant downdrafts that inhibit further growth of thermals (Seifert and Heus, 2013; Seifert et al., 2015; Seigel, 2014). At the boundaries of these open cells, convective formation is promoted due to the moistening of downdrafts (Seifert and Heus, 2013) and mechanical lifting due to the convergence of cold pool outflows (Xue et al., 2008).

This dynamics between the formation of cold pools from precipitation and the subsequent formation of clouds manifests as temporal oscillations; the cloud field goes through a relatively weak convective phase, until multiple downdrafts from the cold pools collide into a convergence zone, where convective growth begins anew. For the mesoscale marine boundary- layer

stratocumulus clouds, the spatial organization of precipitation is found to be important in promoting subsequent cloud formation and the evolution of open cell convection (Feingold et al., 2010; Koren and Feingold, 2013; Wang and Feingold, 2009; Yamaguchi and Feingold, 2015). High-resolution large-eddy simulations have confirmed the formation of cold pools as the main mechanism that drives organized marine stratocumulus convection, which corresponds well to long-term satellite measurements (Bretherton and Blossey, 2017; Seifert and Heus, 2013; Zuidema et al., 2012).

The temporal oscillation has also been observed in modelling studies of precipitating cumulus convection (Dagan et al., 2018; Feingold et al., 2017). For both shallow and deep clouds, the dominant mechanism that drives this oscillatory evolution is found to be the formation of cold pools due to evaporative cooling from precipitation in the sub-cloud layer (Seifert and Heus, 2013; Yano and Plant, 2012; Tompkins, 2001). This mechanism is referred to as the recharge-discharge cycle of thermodynamic instability by Dagan et al. (2018), motivated by Bladé and Hartmann (1993), where the evaporative cooling due to precipitation charges instability in the atmosphere, which is discharged by convection.

Precipitation facilitates both the spatial organization and temporal oscillation of the cloud field, and is governed by a number of factors including cloud microphysics and cloud layer depth. Aerosols, acting as cloud condensation nuclei (CCN), can influence the cloud microphysics by enhancing the cloud droplet number concentration but suppress droplet growth (Twomey, 1974), and large-eddy simulation (LES) studies have shown that when the aerosol concentration is increased, the cloud layer deepens, which then affects rain formation (Dagan et al., 2017; Seifert et al., 2015). Furthermore, modelling studies have shown that an increase in aerosol concentration influences both the amount and the timing of precipitation; in a polluted environment, the efficiency in precipitation formation is reduced, and as a result, rain formation is suppressed and delayed (Dagan et al., 2018; Seifert et al., 2015; Yamaguchi et al., 2019)".

Furthermore, to ensure that we have a clear picture of how precipitation influences the cloud size distribution, We performed two new model runs based on the CGILS case with the precipitation turned on and off. In all cases where precipitation was suppressed (for both BOMEX and CGILS), no oscillatory behaviour was observed. The corresponding paragraph has now been moved to line 528, which now reads: "To further examine the effect of precipitation in the recharge-discharge cycle, we performed a follow-up simulation of the CGILS case with precipitation manually turned off in the microphysics scheme. Likewise, we have tested the GP regression method to the mass flux time-series in weakly-precipitating shallow convection during BOMEX (Holland and Rasmusson, 1972) case. In both cases, the GP

[Figure]

Figure 1: Mean posterior distributions from the GP regression for vertical mass flux $\overline{M}$ (orange) and the observed time-series of vertical mass flux (blue) from the CGILS case where precipitation has been suppressed.

regression method described in Section 2 failed to converge towards a single periodicity". The result of the non-precipitating CGILS case run is shown in Figure 1 below.

As the new paragraph suggests, we have performed another simulation of the CGILS case with precipitation manually turned off. Figure 1 shows the time-series of vertical mass flux $\overline{M}$ and the mean posterior distribution from our GP regression model. As shown here, the GP regression model fails to pinpoint the oscillatory behaviour, as the resulting posterior distribution (orange line) seems to consist of more than one period. We have tested different priors, but when there is no precipitation, the GP regression model fails to model the observed data points with a single periodicity.

2. The imbalance between the more technical parts of the paper and the interesting discussion that starts on pg 16 could be corrected a bit. I felt that there were missed opportunities on the discussion to dig into the boundary layer physics. Examples: lines 412-414, lines 415-417, but I think there is much more that can be said in Sections 3.2, 3.3, and 3,4. The figures require some work since one has to mentally superimpose plots of b and M, or pick off peaks and troughs in different plots to see that they are out of phase. Also, because b is normalized, it should be made clearer that a smaller b is a more negative b, i.e. a larger fraction of small clouds. The plots contain important information but reference to them is too cryptic in my opinion.

[Figure]

Figure 2: Mean posterior distributions from the GP regression for the slope $b$ of the cloud size distribution (blue), cloud fraction $f_c$ (orange) and average vertical mass flux $\overline{M}$.

We have come up with an additional figure that shows how the oscillations occur in different phases (see Figure 2), which will be added to the revised manuscript. Figure 2 makes it clear that the average vertical mass flux $\overline{M}$ varies inversely to the slope of the cloud size distribution. We have also added an additional paragraph, which reads (line 512): "Figure 15 shows the oscillatory evolutions of the mean posterior distributions for $\tilde{b}$, $f_c$ and $\overline{M}$, which gives more insight about how the marine boundary-layer cumulus clouds evolve over time in a high-resolution model. When there is a relative abundance of larger clouds, the normalized slope $\tilde{b}$ of the cloud size distribution and cloud fraction $f_c$ become larger, which corresponds to a less negative (less steep) slope $b$ of the cloud size distribution. Hence, the changes in cloud fraction $f_c$ is correlated to the number of large clouds; that is, the number of large clouds (mostly observed as anvil-like structures near the cloud layer top) determines how much of the model domain is covered by clouds".

We agree that normalization makes it difficult to see what $b$ represents. We tried to relate the normalized $b$ values to the actual slope. For example, line 367 now reads: "The normalization, however, makes it more difficult to see that the time-series of the slope $b(t)$ of the cloud size distribution has a negative sign (see Figure 6). A small value in the normalized slope $\tilde{b}$ indicates a more negative slope, or a steeper slope where there is a relative abundance of smaller clouds. On the other hand, a large value in $\tilde{b}$ represents a less negative slope where there is a relative abundance of larger clouds".

We have also added more discussions about the boundary layer physics, especially the factors

that disrupt the oscillation. For example, we discuss the deviations in $b(t)$ at line 419 reads: "Based on Figure 9 at around $t = 4$ hours, the cloud field is expected to go towards a phase of relatively weak convection where there is a relative abundance of smaller clouds. Small normalized values correspond to more negative slopes of the cloud size distribution. However, upon inspecting the evolution of the cloud field during this time, large structures form at the top of the cloud layer (light grey regions in Figure 11) as a result of strong convective activity, which persist until $t = 5$ hours into the simulation. Once the large, thin layer of clouds at the top dissipates, the deviation the observed time-series of $b$ becomes much smaller. During this time, as shown in Figure 11, the cloud field is dominated by the growth of small clouds. However, a thin layer of clouds persist at the top of the cloud layer, which skews the slope $b$ of the cloud size distribution".

3. What about the possibility of a charge (production of instability) - discharge (consumption of instability) as a driver of oscillations. Is this how non-precipitating systems differ from precipitating systems?

We believe this can be answered by our response to the first point. We re-ran the simulations to confirm that, as the reviewer noted, the time-series of vertical mass flux and cloud fraction in non-precipitating systems clearly differ from precipitating systems. We would like to apologize again for the confusion concerning the BOMEX case, which we have confirmed to behave like non-precipitating CGILS case with no apparent oscillatory behaviour.

4. Oscillations in M lead the oscillations in CF by 45-50 minutes. This made me wonder which size clouds contribute most to CF, which of course depends on b. I also wondered how detrainment at cloud top contributes to this (see Fig. 1). Could you strengthen this analysis and discussion?

We have added a discussion on which clouds contribute most to cloud fraction. Line 512 reads: "Figure 15 shows the oscillatory evolutions of the mean posterior distributions for $\tilde{b}$, $f_c$ and $\overline{M}$, which gives more insight about how the marine boundary-layer cumulus clouds evolve over time in a high-resolution model. When there is a relative abundance of larger clouds, the normalized slope $\tilde{b}$ of the cloud size distribution and cloud fraction $f_c$ become larger, which corresponds to a less negative (less steep) slope $b$ of the cloud size distribution. Hence, the changes in cloud fraction $f_c$ is correlated to the number of large clouds; that is, the number of large clouds (mostly observed as anvil-like structures near the cloud layer top) determines how much of the model domain is covered by clouds".

5. Dagan et al. (2018) change their domain size and show that oscillations get smoothed out for larger domain sizes - at least for BOMEX. Have you tested

the sensitivity of the periodicity to domain size. One could imagine that small domains introduce higher frequency oscillations because of the periodic boundary conditions. If the oscillations become harder and harder to discern at large domain size, are they really important? The fact that they can be discerned by your GP regression is very nice but they may not have much physical significance in a natural cloud system unless the system experiences strong internal coupling (e.g., open-cell Sc)

That is exactly the aim of this manuscript; that is, although the oscillations become more difficult with increasing domain size, numerical methods can be used to identify the oscillatory behaviour and estimate the uncertainties involved using Bayesian inference. Feingold et al. (2017) also did this on a smaller domain, but had difficulties in identifying the period in the cloud size distribution. It becomes more and more difficult to identify the oscillatory behaviour due to the variability in the time-series as the domain size becomes larger. So the effect of the oscillatory behaviour is simply hidden in a large cloud field, and the variability in the properties of the cloud field can be better determined if we can account for the oscillatory evolution. The variability due to the oscillatory behaviour still exists; we believe that the more we know about the internal dynamics of these large cloud fields, the better we can understand their behaviour.

We added a relevant discussion starting line 592: "We have also observed the spatial organization of shallow convective clouds that disrupts the oscillatory evolution (Figure 12), which can also be seen in studies modelling the boundary-layer cloud field over a smaller domain (Wang and Feingold, 2009). Reducing the size of the model domain may reduce the effect of spatial organization and make it easier to estimate the periodic behaviour of the cloud field using traditional methods. However, given that these factors can manifest even on a smaller domain, a robust method to estimate the periodicity of a noisy, non-stationary time-series is still useful, especially if no smallest, *optimal* domain size exists where the recharge-discharge cycle can be isolated".

> 6. It would be nice to know how the normalized b values translate to actual b values that e.g., a satellite imager would see.

Thank you for pointing this out. We briefly compare this to Neggers et al. (2003) and Brown (1999) (line 159 in the updated manuscript), but we should have brought this up again when we calculate $b$. The corresponding discussion starts at line 192, which reads: "The cloud size distribution C(a) given in Figure 3b represents a normalized probability density function, which will differ from the histograms obtained from observations. Here,

the slope is measured to be roughly $b = -0.65$. We have calculated b for non-normalized values of $C(a)$, and the time-series is found to vary roughly between $-1.4$ and $-1.7$, which corresponds to a range of $-0.7$ to $-0.85$ based on the methods by Neggers et al. (2003), and $-1.7$ to $-1.85$ by Benner and Curry (1998). The measured slopes are slightly smaller in magnitude but comparable to the slope of $b = -1.7$ found in a large-eddy simulation (Neggers et al., 2003) and the slope of $b = -1.98$ from remote sensing observations (Cahalan and Joseph, 1989; Benner and Curry, 1998)".

7. Line 132: could you help with physical meaning of bandwidth h?

We have re-written the following paragraph in hopes of making it clearer. The paragraph starting at line 157 now reads: "Each cloud size sample is added to a distribution not as a single point of observation, but a probability distribution based on a Gaussian distribution. It can be considered as an uncertainty in the measurement; that is, each cloud sample is considered to be a Gaussian probability distribution whose width is defined by the bandwidth $h$".

8. Line 102, $q_l > 0$ is a very low threshold, unless your cloud edges are very sharp. Does $q_l > 0.01$ g/kg change the picture?

$q_l > 0.01$ g/kg is typically used for large-scale models, but not widely for large-eddy simulations. We have tested different conditional criteria for cloud sampling in the past, but have not found any differences in statistical distributions, likely thanks to the sheer number of cloud samples taken from the model run.

---

## Author Comment (AC5)

**Response to comments for "Quantifying the Oscillatory Evolution of Simulated Boundary-Layer Cloud Fields Using Gaussian Process Regression"**

Loren Oh*

February 12, 2025

We sincerely appreciate the constructive feedback for the submitted manuscript. We have made extensive changes to the manuscript based on the reviews, and performed additional model runs to reflect the points raised by the reviewers. The following document summarizes those points, and the changes we've made.

The line numbers in the responses refer to those in the revised manuscript.

**Responses to Comments**

**Major Points**

> 1a. I see spectral analysis as the gold standard for identifying periodic signals in a dataset - if the dataset has a signal at a certain wavelength, it *must* show up in a spectral analysis. I also disagree that spectral analysis is particularly sensitive to noise - noise usually shows up in a power spectrum as a lower limit to spectral power across a range of frequencies. It is true that this lower limit can be so large that it swallows your signal, but there are 2 peaks in Fig. 5 that rise above this floor. In this context it is hard to square the results of your GPM against what I see in your power spectrum or even via visual inspection of your time series.

Thanks for pointing this out. We realized that some of the steps we have taken were left unjustified in the original draft. One of the things we observed from the time-series is that it is very noisy and non-stationary, which makes DFT less informative. The relevant observation
* * *
*Corresponding author: loh@eoas.ubc.ca

at line 208 now reads: "We are interested in determining whether the fluctuations in the time-series of the cloud size distribution in Figure 4 are consistent with a periodic behaviour. The oscillatory evolution in b is not immediately obvious in Figure 4, and performing an Augmented Dickey-Fuller (ADF; Dickey and Fuller, 1979; Hamilton, 2020) test reveals that the time-series is non-stationary. That is, the observed time-series $b(t)$ cannot be assumed to be stationary, which would have a unit root. We would like to quantify the extent to which the time-series is consistent with earlier studies regarding oscillations in the cloud size distribution. In the following section, we follow Feingold et al. (2017) and use Fourier spectral analysis to identify the underlying periodic behaviour in the observed time-series".

In the presence of noise and non-stationarity, it comes down to the problem of interpretation. On this front, we encountered three issues with both DFT and ACF, which is that the observed data is quite noisy (which may or may not behave as white noise), that the time-series is non-stationary and that the signals are found in the low-frequency regime.

To show that these factors make interpretation of the spectral analysis more difficult, we defined the *noise floor* to be the magnitude of the spectral power for Gaussian noise using a chi-squared distribution. We have revised the manuscript to include more details about this particular choice, and the relevant paragraph at line 226 now reads: "Figure 5 shows the estimate of the power spectral density using the periodogram (blue) and the 95% confidence interval (red) obtained from the noisy time-series b(t), plotted as a function of period T(k) = N/k. The 95% confidence interval defines the threshold that separates oscillatory signals from noise, against the null hypothesis that all signals in the periodogram are Gaussian noise, and is based on a chi-squared distribution with 2 degrees of freedom (Panofsky and Brier, 1958)".

The goal here was to see if the traditional spectral analysis could identify the 90-minute period with enough confidence (i.e. 95% confidence that the signal is not white noise). From what I can tell from the spectral analysis, the time-series indeed contains large variability for the spectral analysis to be effective.

> 1b. I'm pretty sure all you are doing is effectively band-pass filtering the data by looking only for periods between 5 and 150 minutes which throws out the large signal at 230 minutes and causes the secondary peak at 100 minutes in Fig 5 to rise to the level of statistical significance because so much of the rest of the power spectrum has been thrown away. I could be wrong about this, but I would need to see my hypothesis proven wrong before I'd be comfortable accepting this paper.

[Figure]

Figure 1: Power spectral density from FFT, with periods larger than 200 minutes filtered out.

We understand the concern, but the choice of the prior does not exclude the signal at longer periods, unlike the DFT. First of all, band-pass filtering the Fourier series still does not yield a strong enough signal given the noise and/or the variability within the periodicity (see Figure 1).

The choice of the prior (periodicity) in the GP regression method, on the other hand, is not meant to be (and is not) deterministic. In most cases, the GP model does not get stuck in local minima. We still tested a range of periods simply to demonstrate that our model does not get stuck in local minima; the GP model converges to the 90-minute periodicity given enough time.

We should definitely have explained why we ignored the 230-minute period. The relevant discussion at line 233 now reads: "There are two prominent periods on the periodogram, one at T = 100 minutes and the other at T = 233 minutes, but only the latter signal is above the 95% confidence interval. Both signals have periods longer than the 80-minute period observed by Feingold et al. (2017) and no significant signals can be found at shorter periods. The signals are found in the low-frequency regime, and because of that, the frequency bin width is too large to pinpoint the exact period from the periodigram, especially for T = 233 minutes. This period is also much longer than the expected oscillatory behaviour (Dagan et al., 2017; Yamaguchi et al., 2019), and is possible that it is a harmonic of a fundamental frequency hidden by the noise and non-stationarity.

We have also tested other methods to estimate power spectral density, such as the circular autocorrelation function (ACF; Parzen, 1962b), but (partial) autocorrelation function of the

[Figure]

Figure 2: Normalized time-series of slope b of the cloud size distribution C(a) (blue line), compared to the the mean posterior from the periodic GP model (orange line), representing the 238-minute oscillation.

observed time-series b(t) decreases slowly over time, and no significant lag can be found. The presence of noise as well as the non-stationary nature of the time-series makes it difficult to examine the behaviour of the time-series".

Still, we can definitely use our GP model to see what the low-frequency oscillation looks like. We used a 230-minute period as the prior, and let the GP model converge slightly (by stopping the gradient descent prematurely) to an oscillation with a period of 238 minutes. See Figure 2; it is clear that the longer period ignores much of the variability in the measured time-series. We have performed a number of trials with longer periods, but found no improvements. The manuscript now mentions this at line

> 1c. I think my simple story in 3b is getting lost in a maelstrom of math. Einstein once said that a model should be as simple as possible, but no simpler. In that context, why do you perform your analysis on the derivative of the signal rather than the signal itself (when there is no trend in the data to get rid of and subtracting the linear regression would be simpler if there were)? Why do you need GPM instead of applying a digital filter? I'd like to see clear justification for each step towards complexity you take

We believe that the reviewer meant to say point 1b.

This must be the third thing that had to be justified, but was missing from the original draft. Thank you for pointing this out again. As we have noted in 1a, the observed time-series is

both noisy and non-stationary. And we have found that taking the derivative of $b$ normalizes and isolates the periodic component, which is indicative of oscillatory component. The relevant discussion starts at line 340, which reads: "The smoothly modelled distribution $\tilde{b}(t)$ from the mean GP posterior distribution $b(t)$ in Figure 6 corresponds well to the observed time-series, showing the oscillatory behaviour within the noisy observation with a period $T = 95$ minutes. In this particular case, the initial regression attempt yields a good estimate of the hyper-parameters for the observed time-series. However, in situations where a general, long-term trend breaks the quasi-stability assumption, additional steps can to be taken in order to better isolate the oscillatory behaviour of the cloud field, which still remains noisy and non-stationary. The standard practice to account for the non-stationarity is to take the derivative of the time-series $\partial_t \tilde{b}(t) = \partial \tilde{b}(t)/\partial t$. If the oscillation is dominated by a single frequency, the frequency should also characterize the derivative of the oscillation. Given this, we build a GP regression model to estimate the period of the oscillation in $\partial_t \tilde{b}(t)$, which can be seen in Figure 7.

We have found that taking the derivative of $b(t)$ also successfully normalizes the observed time-series, which is useful for statistical analysis. Applying the ADF test (Dickey and Fuller, 1979; Hamilton, 2020) to $\partial_t \tilde{b}(t)$ also confirms that the resulting time-series is now stationary. As shown in Figure 7, the values of $\partial_t \tilde{b}(t)$ varies with zero mean with no obvious trend over time. There are small variations in the amplitude, but we can now drop the SE kernel to account for the variability in the y-axis, and only use the periodic kernel (i.e. $\hat{k} = k_{\mathrm{per}}$) to estimate of the periodicity $T$. This reduction in the number of hyper-parameters is also necessary to perform Bayesian inference, which will be described in more detail in Section 2.7".

> 2. I'm concerned that oscillations in cloud populations might be fairly local in nature such that averaging over a 45 km fetch averages out most of the variability you seek. Thankfully, you can easily address this question (which would be a nice addition to your paper)! In particular, can you repeat your analysis by only analyzing output over subsets of your domain? If so, does the peak frequency or spectral power change as a function of region size? Also, discussion of averaging over a large region occurs around line 330, but I'd like to see more discussion and I'd like to see it show up in the experimental design section.

We performed the analysis again over a (randomly chosen) quarter of the domain. Figure 3 shows the resulting power spectral density. As the reviewer suspected, the signals appear stronger on a smaller domain, which is expected as expanding the size of the domain would introduce more variability in our time-series. Still, the spectral analysis returned multiple

[Figure]

Figure 3: Power spectral density from FFT for a quarter of the domain.

signals. We can see additional signals at 144 and 180 minutes above the 95% confidence bound, which is interesting.

Still, the signals at 180 and 240 minutes are likely harmonics of higher-frequency oscillations, and the bin size for the DFT is too large to determine what the actual period should be. Given that, we used 90 and 144 minutes as priors for our GP model, and tested how accurate they were (see Figures 4 and 5). The former converged to a 95-minute period, which is expected, and the latter converged to a 144-minute period, which is less accurate. So it does not become significantly easier to identify the oscillatory evolution of the cloud field even on a smaller domain, as the variability is still too large for traditional methods.

> 3. When I look at Fig 9, I see 4 very clear waves with 95 min period while the rest of the timeseries doesn't fit this frequency very well at all. This leads me to suggest that your active/quiet hypothesis works well for some chunks of your simulation but something is interrupting the signal for other chunks. Can you watch a movie of your simulation or perform other analysis to figure out what is interrupting your behavior of interest? Finding conditions which disrupt these oscillations would be a very noteworthy contribution!

The revised manuscript introduces quite a bit of discussions on possible causes of those deviations. For example, line 419 reads: "Based on Figure 9 at around $t = 4$ hours, the cloud field is expected to go towards a phase of relatively weak convection where there is a relative abundance of smaller clouds. Small normalized values correspond to more negative slopes of the cloud size distribution. However, upon inspecting the evolution of the cloud

[Figure]

Figure 4: Normalized time-series of slope b of the cloud size distribution C(a) (blue line), compared to the the mean posterior from the periodic GP model (orange line), representing the 95-minute oscillation.

field during this time, large structures form at the top of the cloud layer (light grey regions in Figure 11) as a result of strong convective activity, which persist until $t = 5$ hours into the simulation. Once the large, thin layer of clouds at the top dissipates, the deviation the observed time-series of $b$ becomes much smaller. During this time, as shown in Figure 11, the cloud field is dominated by the growth of small clouds. However, a thin layer of clouds persist at the top of the cloud layer, which skews the slope $b$ of the cloud size distribution.

On the other hand, around 9 hours into the simulation, the cloud field is expected to go towards a phase of relatively strong convection where there is a relative abundance of larger clouds. Visually inspecting the development of the cloud field (Figure 12) suggests that strong convective activities occur in groups, and clouds tend to merge and become much larger, which reduces the relative number of large clouds compared to small ones that are less likely to merge with other clouds or reach the cloud top layer. Because of this, the normalized value of $b$ of the cloud size distribution becomes smaller, indicating a much steeper slope despite the strong convective activity".

We have also added Figures 11 and 12, which are snapshots of the model run to give more insight into the state of the cloud field.

> 4. I'm also worried that the narrow dimension of your bowling-alley domain might be corrupting your results. It seems to me that the narrow horizontal direction would constrain the size of cloud radii. And since your whole paper

[Figure]

Figure 5: Normalized time-series of slope b of the cloud size distribution C(a) (blue line), compared to the the mean posterior from the periodic GP model (orange line), representing the 144-minute oscillation.

> centers around the cloud size distribution, locking the biggest clouds to the domain size seems like it could affect your conclusions. Can you reassure me that this isn't happening? Just providing citations of previous work showing this isn't a concern would be fine.

The model domain is definitely large enough for cloud growth, as if it were not the case, the clouds will definitely grow towards the size of the domain and it becomes obvious that the domain is too small. We have added snapshots from the model run, and while there are no previous studies specifically for the CGILS S6 case, doi.org/10.5194/gmd-6-1261-2013 could be a reference for this.

However, it should be mentioned that we have observed what appears to be the effect of spatial organization (due to the formation of cold pools). If we were to study how cold pools affect this oscillatory behaviour, the domain size might not be large enough. This response relates to comment 6, so we will mention the relevant corrections there.

> 5. around line 105: I don't understand whether you're just taking cloud cross sections at a single level (e.g. 1 km) or whether you consider each vertical level to be an independent cloud. The latter approach could be problematic because clouds at different levels wouldn't be independent samples which would mess up statistical analysis tests. This would also distort interpretation because what we think of as a distribution of cloud sizes could in fact just represent

the vertical structure of a single cloud. Can you clarify what you're doing and address potential concerns if you are using all levels?

We realized that there were some unnecessary discussions on cloud sampling, so we tried to make it clear. Line 128 now reads: "In order to obtain the cloud size distribution, individual clouds need to be sampled conditionally. In this study, horizontally contiguous regions (grid cells) containing condensed liquid water ($q_l > 0$) are considered to be the cloud region. The *size* of a cloud is then defined as the area of the horizontal cross-section, which is the number of grid cells containing condensed liquid water multiplied by the horizontal grid size".

Connected convective columns being statistically correlated is definitely a possible issue, which we were aware of. We initially used 60% of the random samples, but found that the statistical distributions appear to be the same regardless. We address this issue at line 130, which reads: "The size distribution contains multiple horizontal cross-sections of the same cloud. However, randomly sampling 60% of the cloud samples had no impact on the analysis, likely thanks to the sheer number of samples and the shallow nature of the modelled clouds in comparison to the size of the domain".

6. There's a lot of focus on matching the periodicity found in previous studies, but do we know that such periodicity should be spatially and temporally invariant? Recharge/discharge seems like it would be proportional to boundary layer depth and the vigor of turbulent/convective mixing. Timescales might also be different in the different sorts of cloud analyzed by these previous papers.

Thanks for the suggestion. This was brought up a couple times in the review, and we have added a bit of discussion on factors that can influence this oscillation, namely aerosol concentration and spatial organization as well as domain size. For example, line 592 now reads "We have also observed the spatial organization of shallow convective clouds that disrupts the oscillatory evolution (Figure 12), which can also be seen in studies modelling the boundary-layer cloud field over a smaller domain (Wang and Feingold, 2009). Reducing the size of the model domain may reduce the effect of spatial organization and make it easier to estimate the periodic behaviour of the cloud field using traditional methods. However, given that these factors can manifest even on a smaller domain, a robust method to estimate the periodicity of a noisy, non-stationary time-series is still useful, especially if no smallest, *optimal* domain size exists where the recharge-discharge cycle can be isolated".

[Figure]

Figure 6: Mean posterior distributions from the GP regression for the slope $b$ of the cloud size distribution (blue), cloud fraction $f_c$ (orange) and average vertical mass flux $\mathcal{M}$.

**Minor Points**

1. It would be nice to see the timeseries of cloud size distribution, mass flux, and cloud fraction all on a single graphic. The degree of correlation between variables is difficult to see by flipping between graphics. A lead/lag correlation analysis would be a nice way to show the relationships between these fields.

Multiple people pointed this out, so to address this point we have come up with an additional figure (see Figure 6 here) showing the oscillations in the three variables.

This has been added as Figure 15 with relevant discussions.

2. A cartoon explaining the relationship between cloud size, mass flux, cloud fraction, and precipitation in terms of hypothesized regional life cycle might be useful for driving home what you're seeing.

This has more to do with the dynamics between mass flux and precipitation, and we thought it would be made redundant as it has been shown by Dagan et al. (2018). We would like to perform a follow-up study that looks into the changes in vertical mass flux distributions, and we will consider making a figure on the relationship once we know more about the mechanism.

3. I really like Figure 1! It does a nice job of grounding the paper in reality. It might be even more powerful if you showed snapshots like this for both a very active and very quiescent period.

Thank you. We added more discussions on factors that disrupt the oscillatory behaviour (based on Figure 9), and added two more snapshots (Figure 11 and 12) to visualize the state of the cloud field and gain more insight into what is causing the deviations.

> 4. The KDE gives you a non-parametric PDF which you can use to get the slope you need...so why do you introduce a power-law distribution? I suspect all you want is to compute the linear regression of the nonparametric KDE PDF in log-log space (after chopping off the nonlinear bit on the left-hand side of the curve in e.g. Fig 3). I suspect I'm complaining about your wording rather than what you've actually done.

There were requests of comparing our result with previous studies (specifically satellite observations). Line 197 reads: "Here, the slope is measured to be roughly $b \approx -0.65$. We have calculated $b$ for non-normalized values of $C(a)$, and the time-series is found to vary roughly between $-1.4$ and $-1.7$, which corresponds to a range of $-0.7$ to $-0.85$ based on the methods by Neggers et al. (2003), and $-1.7$ to $-1.85$ by Benner and Curry (1998). The measured slopes are slightly smaller in magnitude but comparable to the slope of $b = -1.7$ found in a large-eddy simulation (Neggers et al., 2003) and the slope of $b = -1.98$ from remote sensing observations (Cahalan and Joseph, 1989; Benner and Curry, 1998)".

> 5. I found Fig 3 to be a bit confusing. I would expect the derivative of the original plot to come below/after the plot it is the derivative of. It also took me a while to see the delta in front of the ylabel in panel a, which added to my confusion. Further, it would be reassuring to see several snapshots in time to make sure your log-log linearity assumption is justified in general instead of just in this test case. Alternatively, you could plot RMSE of fit (or something like that) for your linear regression as a function of time.

We found that the panel a in Figure 3 is actually never mentioned in writing. We apologize for the confusion. Line 184 now reads: "A decision tree regression algorithm (Breiman et al., 1984) is used to divide the distribution into two parts by limiting the maximum number of possible branches to two, corresponding to the portion of the distribution with relatively constant slope and the rest of the distribution. This is effectively done by fitting a simple piecewise-constant function $\hat{C}(a)$ to the derivative of $C(a)$, as shown in Figure 3a, where the breakpoint $\hat{a}$ which minimizes the error between the distribution $C(a)$ and $\hat{C}(a)$. Here, the error is defined as the mean square error".

**Typos**

Thanks for catching these. We will fix them in the revised manuscript.